# The Ragulator complex and lysosomal calcium release are crucial for cell migration

Tatsunori Jo[1,2,3] ⓘ, Kohei Tsujimoto[1,2,3] ⓘ, Takeshi Nakatani[1,4], Daiki Nagira[1], Yutaka Muto[1,2,3], Takehiro Hirayama[1,2,3], Hachiro Konaka[1,5], Masato Okada[6], Hyota Takamatsu[1,7] ⓘ, Atsushi Kumanogoh[1,2,3,8,9,10,11] ⓘ

Immune cells migrate via actomyosin contractility mediated by myosin IIA activation, wherein the lysosomal Ragulator complex–MPRIP interaction is crucial. However, the precise mechanism underlying lysosome-mediated myosin IIA activation has not been elucidated. Here, we found that calcium efflux from the lysosomal TRPML1 channel promotes leukocyte trafficking by enhancing the interaction between the Ragulator complex and MPRIP. Disrupting the lysosome-anchoring site of Lamtor1 impaired the localization of the Ragulator complex to lysosomes, diminishing the TRPML1-mediated leukocyte migration and interaction between Lamtor1 and MPRIP. Furthermore, ouabain, a cardiac glycoside, dissociated Lamtor1 from lysosomes, inhibiting the interaction between the Ragulator complex and myosin IIA activation, thereby suppressing cell migration. Therapeutically, ouabain ameliorated the severity of MSU-induced gouty arthritis and LPS-induced lung injury in mice by inhibiting leukocyte infiltration. Overall, lysosomes facilitate the interaction between the Ragulator complex and MPRIP by supplying calcium ions through TRPML1 channels, thereby activating myosin IIA and promoting leukocyte migration.

## Introduction

Leukocyte migration is a highly regulated process involving several steps including rolling, adhesion, crawling, and transmigration through the endothelium and interstitial tissues (Muller, 2003; Ley et al, 2007). During the initial stages of migration, leukocytes tether to and roll along the vessel wall using selectins and integrins as adhesion molecules and transmembrane linkers (Ley et al, 2007). Once leukocytes cross the endothelial layer to the ECM, myosin II accumulates at the rear of the cell. Subsequently, the myosin light chain (MLC) is phosphorylated to induce actomyosin contraction, leading to relatively rapid movement (Lammermann et al, 2008; Vicente-Manzanares et al, 2009; Paluch et al, 2016). MLC phosphorylation is dynamically regulated by the MLC kinase (MLCK) and MLC phosphatase (MLCP). MLCK is a calcium/calmodulin-dependent serine/threonine kinase, wherein chemokines activate inositol trisphosphate (IP3) receptors (Berridge, 2009) to release Ca2+ ions from the ER, inducing a conformational change in calmodulin (Devreotes & Zigmond, 1988; Mandeville & Maxfield, 1996). MLCK activation enhances myosin IIA activity. On the contrary, MLCP, a heterotrimer consisting of M20 and myosin phosphatase target subunit 1 (MYPT1), the catalytic subunit of the type I protein serine/threonine phosphatase family (PP1c subunit) (Vicente-Manzanares et al, 2009), is tethered to the actin bundle by interacting with the actin-binding protein myosin phosphatase Rho-interacting protein (MPRIP) (Surks et al, 2003). MPRIP and MYPT1 interaction sustains the dephosphorylation of MLC, thereby reducing myosin IIA activity (Kimura et al, 1996).

During cell migration, lysosomes move toward the cell periphery (Matteoni & Kreis, 1987), wherein peripheral lysosomes have also been reported to allow uropod to detach from the ECM by releasing proteases (Colvin et al, 2010) and modulating the stability of focal adhesions during cell migration (Paluch et al, 2016). In addition, peripheral lysosomes have been suggested to activate myosin IIA and maintain the actin bundle at the uropod via the lysosomal calcium channel TRPML1 (Bretou et al, 2017). We recently reported that the lysosomal Ragulator complex, a nutrient-sensing hub that anchors mTORC1 (Sancak et al, 2010; Saxton & Sabatini, 2017), is implicated in actomyosin-mediated immune cell migration independent of mTORC1. During cell migration, lysosomes move to the rear of the cell, where the Ragulator complex, a pentameric

[1]Department of Respiratory Medicine and Clinical Immunology, Graduate School of Medicine, Osaka University, Osaka, Japan   [2]Department of Immunopathology, WPI, Immunology Frontier Research Center (iFReC), Osaka University, Osaka, Japan   [3]The Japan Science and Technology – Core Research for Evolutional Science and Technology (JST–CREST), Osaka University, Osaka, Japan   [4]Department of Respiratory Medicine, Osaka Keisatsu Hospital, Osaka, Japan   [5]Department of Internal Medicine, Nippon Life Hospital, Osaka, Japan   [6]Department of Oncogene Research, Research Institute for Microbial Diseases, Osaka University, Osaka, Japan   [7]Department of Clinical Research Center, National Hospital Organization Osaka Minami Medical Center, Osaka, Japan   [8]Integrated Frontier Research for Medical Science Division, Institute for Open and Transdisciplinary Research Initiatives, Osaka University, Osaka, Japan   [9]Japan Agency for Medical Research and Development – Core Research for Evolutional Science and Technology (AMED–CREST), Osaka University, Osaka, Japan   [10]Center for Advanced Modalities and DDS (CAMaD), Osaka University, Osaka, Japan   [11]Center for Infectious Disease for Education and Research (CiDER), Osaka University, Osaka, Japan

Correspondence: thyota@imed3.med.osaka-u.ac.jp; kumanogo@imed3.med.osaka-u.ac.jp

complex consisting of Lamtor1/p18 surrounding Lamtor2/p14, Lamtor3/MP1, Lamtor4/p10, and Lamtor5/HBXIP (Yonehara et al, 2017), interacts with MPRIP to interfere its interaction with MYPT1, enhancing the phosphorylation of MLC by turning off MLCP activity, thereby propelling the cell forward (Nakatani et al, 2021). However, the role of lysosomes in myosin IIA activation via the Ragulator complex during leukocyte migration has not yet been elucidated.

Here, we show that lysosomal calcium efflux via TRPML1 facilitates the interaction between the Ragulator complex and MPRIP, promoting actomyosin-mediated immune cell motility. In addition, we identified that ouabain, a cardiac glycoside, alleviates cell motility and inflammation by interfering with the Ca2+-mediated interaction of the Ragulator complex with MPRIP by preventing the localization of the Ragulator complex to lysosomes.

## Results

### Calcium efflux from lysosomes via TRPML1 enhances Lamtor1–MPRIP interaction

To investigate the effect of calcium ions on Lamtor1 and MPRIP interactions, immunoprecipitation assays were performed using HEK293T cells expressing Lamtor1-Flag and MPRIP-V5 at different Ca2+ concentrations. We found an enhancement of the Lamtor1–MPRIP interaction with increasing Ca2+ levels, an effect that was canceled by the addition of EDTA post-treatment (Fig 1A). Given that TRPML1 is crucial for regulating lysosomal calcium efflux and functions (Xu & Ren, 2015), we assessed Lamtor1 and TRPML1 localization through immunohistochemical staining of THP1 cells, a human monocytic cell line, revealing their close proximity (Fig 1B). Immunoprecipitation assays in HEK293T cells expressing Lamtor1-Flag and TRPML1-V5 confirmed their direct interaction (Fig 1C). We next examined the influence of TRPML1-mediated Ca2+ efflux on the Lamtor1–MPRIP interaction using MLSA-1, a TRPML1 agonist, or MLSI-3, a TRPML1 antagonist. In the presence of MLSA-1, enhanced immunoprecipitation between Lamtor1 and MPRIP was observed in HEK293T cells expressing Lamtor1-Flag and MPRIP-V5 (Fig 1D). We also evaluated these interactions using NanoBRET assays in HEK293T cells transfected with Lamtor1 tagged with a NanoLuc acceptor at the amino terminus (Lamtor1-NLF-C) and MPRIP$_{1-539}$ with Halo-Tag donor at the amino terminus (MPRIP$_{1-539}$-HTC). We confirmed an increased BRET intensity with MLSA-1 administration and decreased intensity with MLSI-3 (Fig 1E). On the contrary, an enhancement of the interaction between Lamtor1 and MPRIP was not observed when cells were treated with A23187, a calcium ionophore that increases overall cellular calcium levels (Fig S1A). These results suggest that lysosomal calcium release is essential in these interactions. We then evaluated the effect of lysosomal Ca2+ release on immune cell migration using THP1 cells. First, we conducted the adhesion assay of THP1 cells to vascular cell adhesion protein 1 (VCAM-1)–coated dishes. The adhesion of THP1 cells was not changed regardless of the

presence of MLSA-1 or MLSI-3 (Fig S1B). Then, we conducted the Transwell assay to evaluate the cell migration. MLSA-1 enhanced THP1 cell migration in response to CCL2 treatment, whereas MLSI-3 suppressed it (Fig 1F). In addition, the phosphorylation of MLC was increased when THP1 cells were treated with MLSA-1 and decreased with MLSI-3 (Fig 1G). We next evaluated the cell morphology of BMDCs during migration within 3D collagen matrices in the presence of MLSA-1, MLSI-3, and blebbistatin, a myosin II ATPase inhibitor. MLSA-1 enhanced the velocity of cells, whereas MLSI-3 and blebbistatin decreased it. Notably, both MLSI-3–treated cells and blebbistatin-treated cells exhibited cell body elongation, but MLSI-3–treated cells elongated with multiple branches, whereas blebbistatin-treated cells elongated without branching (Fig 1H, Video 1, Video 2, Video 3, and Video 4). In addition, lysosomes were distributed in the trailing edge in BMDCs migrating within 3D collagen matrices even in the presence of MLSI-3, although TRPML1 has been shown to affect lysosomal trafficking (Fig S1C, Video 5 and Video 6). These findings demonstrate that calcium efflux from lysosomes via TRPML1 is crucial in the actomyosin-mediated cell migration via enhancing the Lamtor1–MPRIP interaction.

### Localization of Lamtor1 to lysosomes is essential for Lamtor1–MPRIP interactions and cell migration

Lamtor1, a crucial anchor for the Ragulator complex on lysosomal membranes, is selectively localized to the lipid rafts of late endosomes through its N-terminal structure. The Lamtor1-G2A variant has its N-terminal G2 region replaced by alanine and is distributed in the cytoplasm rather than being colocalized with lysosomes because of altered anchoring (Nada et al, 2009). We thus evaluated the significance of lysosomal localization of the Ragulator complex in immune cell migration by reconstituting Lamtor1-KO-THP1 cells with a Lamtor1-G2A variant (Lamtor1-KO-G2A-THP1 cells). We confirmed impaired localization of Lamtor1-G2A to lysosomes (Fig S2A) and decreased Lamtor1-G2A protein in lysosome-enriched fractions (Fig S2B). Using the Lamtor1-G2A construct, we firstly investigated the interaction between Lamtor1-G2A-Flag and MPRIP-V5 by coimmunoprecipitation assays, revealing their reduced interaction (Fig 2A). We then evaluated the chemotaxis of Lamtor1-KO-G2A-THP1 cells upon CCL2 treatment and found decreased migration compared with that of Lamtor1-Full-THP1 cells (Fig 2B). We then evaluated myosin IIA activity and observed decreased MLC phosphorylation in Lamtor1-KO-G2A-THP1 cells (Fig 2C). Notably, MLSA-1–enhanced interaction between Lamtor1 and MPRIP was diminished in Lamtor1-G2A-expressing HEK293T cells (Fig 2D). In addition, the enhanced migration and increased phosphorylated MLC of WT-THP1 cells caused by MLSA-1 were abrogated in Lamtor1-KO-G2A-THP1 cells (Figs 2E and 3F). Furthermore, we performed RNA sequencing of Lamtor1-G2A and confirmed that the variant does not significantly alter the expression of genes related to cell migration (Fig S2C, Supplemental Data 1). These results revealed that Lamtor1-G2A affects cell migration by diminishing the TRPML1-mediated protein–protein interaction between the Ragulator complex and MPRIP. This emphasizes

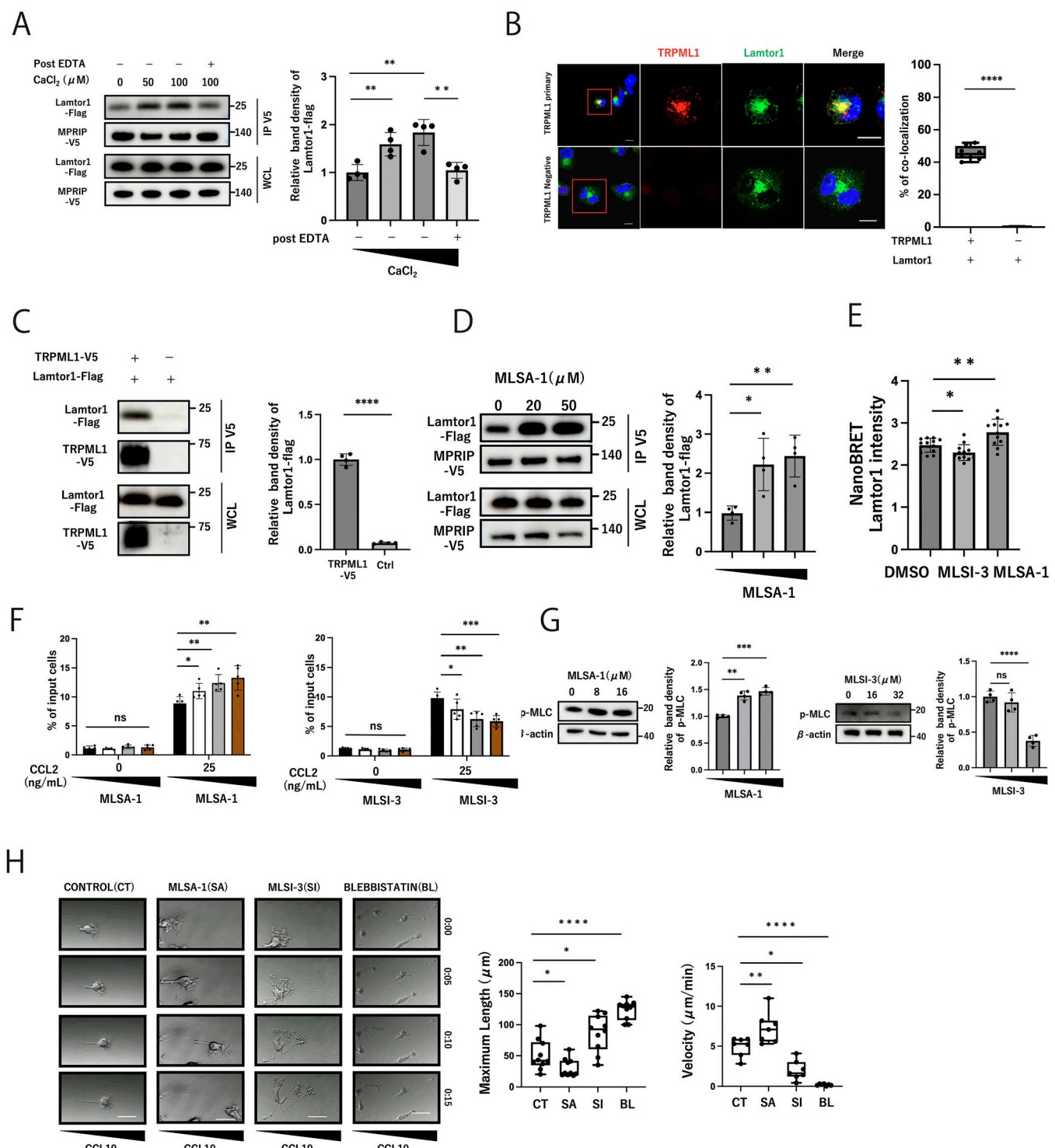

**Figure 1. Calcium efflux from lysosomes via TRPML1 enhances Lamtor1–MPRIP interaction.**
**(A)** Effect of calcium on the interaction between Lamtor1 and MPRIP. The V5-tagged MPRIP expression vector was transfected into Lamtor1-Flag-expressing HEK293T cells. Cells were lysed, and EDTA (3 mM) was added to the lysate for 6 h, and subsequently, CaCl$_2$ (0, 50, and 100 $\mu$M) was added to the lysate at 4°C for 12 h. Immunoprecipitation was performed using an anti-V5 antibody and detected by Western blotting using anti-V5 and anti-FLAG antibodies. N = 4. **(B)** Localization of Lamtor1 and TRPML1 in WT-THP1 cells was evaluated using confocal microscopy with anti-TRPML1 and anti-Lamtor1 antibodies. For the control group, only secondary antibodies for TRPML1 were used to compare with the experimental group. A representative image of TRPML1 (red) and Lamtor1 (green) is shown; scale bar, 10 $\mu$m. N = 8.
**(C)** Immunoprecipitation assay between TRPML1 and Lamtor1. The V5-tagged TRPML1 expression vector was transfected into Lamtor1-Flag-expressing HEK293T cells. Cells were lysed and immunoprecipitated with an anti-V5 antibody, detected by Western blotting using anti-V5 and anti-FLAG antibodies. N = 4. **(D)** Effect of MLSA-1 on the interaction between Lamtor1 and MPRIP. The V5-tagged MPRIP expression vector was transfected into a Lamtor1-Flag-expressed HEK293T cell. After 6 h of treatment with

that the lysosomal localization of the Ragulator complex is essential in cell migration.

## Ouabain inhibits the Lamtor1–MPRIP interaction by preventing the localization of Lamtor1 to lysosomes

We then performed NanoBRET screening to identify substances that could interrupt the Lamtor1–MPRIP interaction. Before NanoBRET screening, we first identified the best combination of the Ragulator complex and MPRIP components and selected Lamtor2-NLF-C and MPRIP$_{1-539}$-HTC because of its highest BRET intensity (Fig S3A–C). Then, we performed NanoBRET screening with 319 phytochemical and 1,112 chemical compounds from the Prestwick Chemical Library comprised of already clinically approved substances (Supplemental Data 2). Twelve phytochemical and three chemical compounds exhibited a reduction in the signal. After excluding compounds that showed reduced nanoluciferase signals because of cytotoxicity, five phytochemical and three chemical compounds were listed as inhibitors of the Ragulator complex and MPRIP interaction. Interestingly, all the compounds were classified as cardiac glycosides (Fig 3A). Among them, we selected ouabain because it has high binding affinity to the α1 isoform of Na K-ATPase, which is predominantly expressed in immune cells, and lower binding affinity to α2 and α3 isoforms, which are expressed in cardiac and muscle cells (Katz et al, 2010). We first examined the effects on Lamtor1, MPRIP, and MYPT1 after ouabain treatment by the immunoprecipitation assay using Lamtor1-Flag–, MPRIP-V5–, and MYPT1-Myc–overexpressed HEK293T cells. A significant reduction in the Lamtor1-Flag and MPRIP-V5 interaction was observed after ouabain treatment (Fig 3B). Consistently, the interaction between MPRIP-V5 and MYPT1-Myc increased upon ouabain treatment owing to the impaired interaction between the Ragulator complex and MPRIP (Fig 3C). We also evaluated the localization of the Ragulator complex after ouabain treatment in THP1 cells. Immunohistochemical staining showed that the colocalization of Lamtor1 and Lamp1 was diminished after ouabain treatment (Fig 3D). In addition, the expression of Lamtor1 in the lysosome-enriched fraction decreased, whereas that in the remnant fraction increased, after ouabain treatment (Fig 3E). We then evaluated whether ouabain could affect the MLSA-1–mediated interaction between Lamtor1 and MPRIP. Immunoprecipitation assay revealed that ouabain attenuated the MLSA-1–mediated interaction between Lamtor1 and MPRIP (Fig 3F). These results suggest that ouabain regulates the interaction between Lamtor1

and MPRIP by delocalizing Lamtor1 from the lysosomal membrane.

Given that the lysosomal Ragulator complex activates mTORC1 and autophagy induction, we assessed LC3 flux and the phosphorylation of pS6K, a downstream target of mTORC1. The phosphorylation of pS6K was not suppressed within the first 12 h of ouabain treatment, but it was suppressed when treatment duration was greater than 24 h (Fig S3D). The expression of LC3B-II was not enhanced by ouabain treatment (Fig S3E). In addition, ouabain did not affect the cell viability within our experimental setting (Fig S3F). Furthermore, the interaction between Lamtor1 and Lamtor2-5 was observed in Lamtor1-G2A-THP1 cells and ouabain treatment (Fig S3G), suggesting that the mTORC1 pathway or complex integrity of the Ragulator complex is not responsible for the decrease in Lamtor1–MPRIP interaction by ouabain. In addition, given that ouabain is a ligand for the ATP1A1 channel, we investigated the involvement of the ATP1A1 channel in the ouabain-mediated localization of the Ragulator complex by generating ATP1A1-knockdown (ATP1A1-KD)-THP1 cells (Fig S3H). Immunohistochemical assay exhibited that WT-THP1 and ATP1A1-KD cells had Lamtor1 expressed in lysosomes at a steady state and were delocalized from lysosomes upon ouabain treatment (Fig 3G). In addition, ATP1A1-KD-THP1 cells did not migrate upon CCL2 stimulation and were suppressed by ouabain treatment (Fig 3H). These results suggest that ouabain delocalizes Lamtor1 from lysosomes independent of ATP1A1, resulting in the attenuation of the TRPML1-mediated interaction between the Ragulator complex and MPRIP.

## Ouabain ameliorates the pathogenic inflammation by inhibiting cell migration

Next, we evaluated the effects of ouabain on immune cell function. Ouabain suppressed CCL2-mediated THP1 cell migration according to the Transwell assay (Fig 4A) and decreased MLC phosphorylation in a dose-dependent manner evaluated by the Western blotting assay (Fig 4B). In addition, ouabain did not affect Rac1 activity, a driver of protrusion at the leading edge (Wittmann et al, 2003), nor RhoA activity, which stimulates actomyosin contractility via Rho-associated protein kinase (ROCK) (Fig 4C). Then, using Lamtor1-KO THP1 (Nakatani et al, 2021) and MPRIP-KO-THP1 cells (Fig S4A), we further examined the effect of ouabain in cell migration. Lamtor1-KO-THP1 cells showed impaired chemotaxis upon CCL2 treatment, regardless of the presence of ouabain, and the enhanced migration induced by CCL2 in MPRIP-KO THP1 cells was not suppressed by

---

MLSA-1, the cells were lysed, immunoprecipitated with an anti-V5 antibody, and detected by Western blotting using anti-V5 and anti-FLAG antibodies. N = 4. **(E)** NanoBRET assay between Lamtor1 and MPRIP by manipulating the TRPML1 channel. Lamtor1 tagged with a NanoLuc acceptor at the amino terminus (Lamtor1-NLF-C) and MPRIP1-539 tagged with a Halo-Tag donor at the amino terminus (MPRIP1-539-HTC) were transfected into HEK293 cells, and the NanoBRET assay was performed using MLSA-1 (TRPML1 agonist) or MLSI-3 (TRPML1 antagonist). N = 12. **(F)** Effect of TRPML1 on cell migration. The chemotaxis in response to CCL2 (25 ng/ml) with different concentrations of MLSA-1 (2, 4, and 8 μM) (left) and MLSI-3 (4, 8, and 16 μM) (right) was evaluated using the Transwell assay system (pore size, 5 μm). N = 5. **(G)** Effects of MLSA-1 and MLSI-3 on MLC phosphorylation. WT-THP1 cells treated with MLSA-1 (0, 8, and 16 μM) and MLSI-3 (0, 16, and 32 μM) each for 6 h were lysed, and Western blotting was performed using an anti-PMLC antibody. N = 5. **(H)** Effect of TRPML1 on motility of dendritic cells (DCs) in response to CCL19 in 3D collagen matrices. Movement of WT BMDCs in response to CCL19 (5 μg/ml) was observed in type I collagen gels (1.5 mg/ml) with MLSA-1 (8 μM), MLSI-3 (16 μM), and blebbistatin (50 μM) in ibidi μ-Slide Chemotaxis. Velocities and maximum length of DCs were determined using ImageJ. Consecutive images of DC locomotion are shown with time (h:min) above each panel. Scale bar, 30 μm N = 10. **(A, B, C, D, E, F, G, H)** Data information: Statistical analyses were performed by a two-sided t test (A, C, D, E, F, G) (means ± s.d) or a two-sided Mann–Whitney U test (B, H) (median; 25th and 75th percentiles; and minimum and maximum of a population excluding outliers). *$P$ < 0.05, **$P$ < 0.01, ***$P$ < 0.001, ****$P$ < 0.0001, NS, not statistically significant.
Source data are available for this figure.

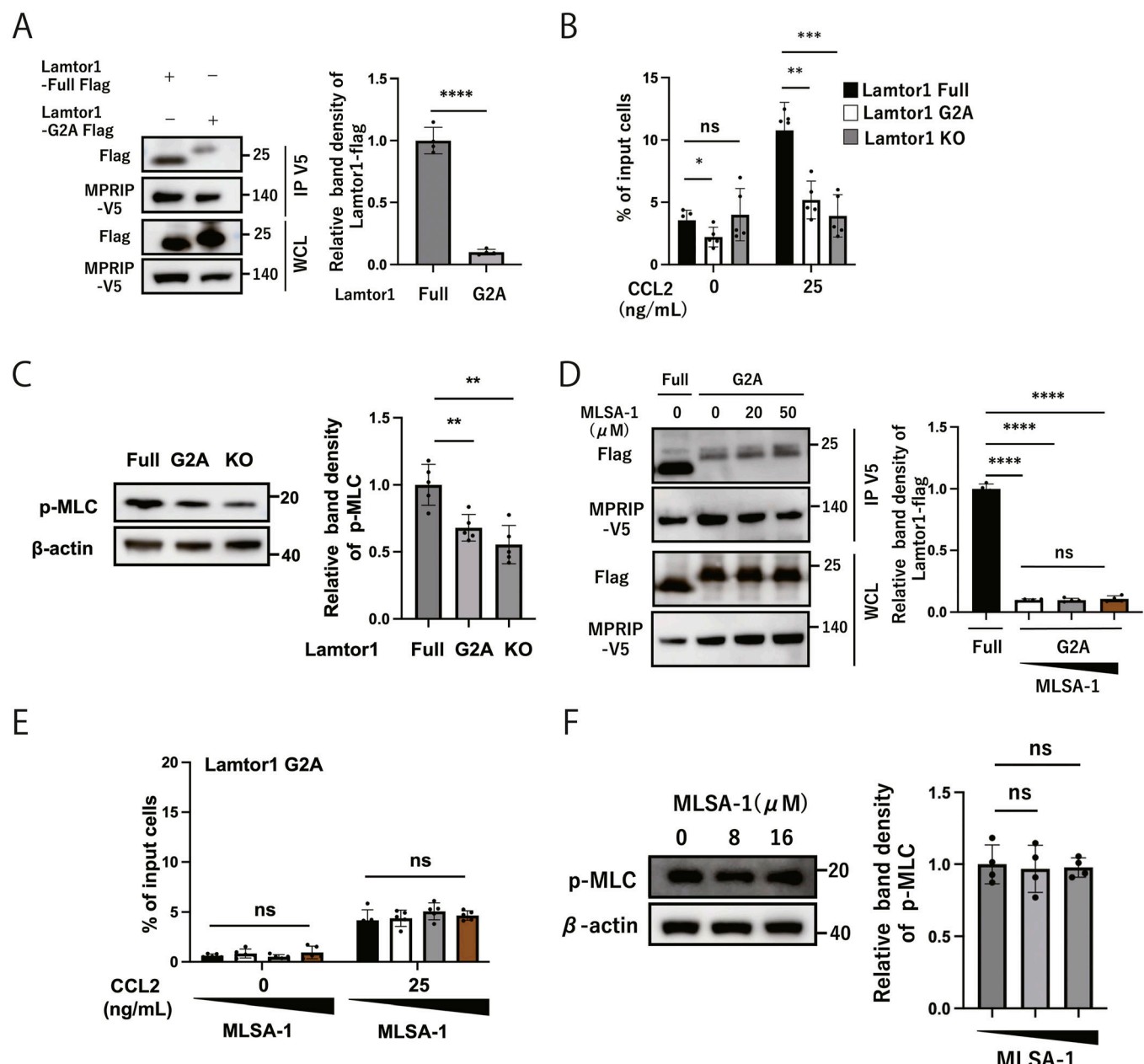

**Figure 2. Localization of Lamtor1 to lysosomes is essential for Lamtor1–MPRIP interactions and cell migrations.**
**(A)** Effect of disrupting lysosomal localization of Lamtor1 on MPRIP interactions. Flag-tagged full-length Lamtor1 vector and Flag-tagged variants of Lamtor1, in which G2 was replaced with alanine (G2A), were transfected to MPRIP-V5-expressing HEK293T cells. Immunoprecipitation was performed using an anti-V5 antibody and detected by Western blotting using anti-V5 and anti-FLAG antibodies. N = 5. **(B)** Effect of disrupting the lysosomal localization of Lamtor1 on cell migration. Chemotaxis of Lamtor1-KO cells, Lamtor1-KO cells reconstituted with full-length Lamtor1 (Lamtor1-KO-Full-THP1 cells), or Lamtor1-G2A (Lamtor1-KO-G2A-THP1 cells) was performed, using the Transwell assay system (pore size, 5 $\mu$m). N = 5. **(C)** Effect of disrupting the lysosomal localization of Lamtor1 on MLC phosphorylation. Lamtor1-KO-THP1, Lamtor1-KO-Full-THP1, and Lamtor1-KO-G2A-THP1 cells were lysed, and MLC phosphorylation was evaluated by Western blotting using an anti-PMLC antibody. N = 5. **(D)** Effects of MLSA-1 on the interaction between Lamtor1-G2A and MPRIP. Flag-tagged Lamtor1-G2A was transfected to MPRIP-V5-expressed HEK293T cells. After 6-h treatment with MLSA-1 (20 and 50 $\mu$M), cells were lysed, and immunoprecipitation was performed with an anti-V5 antibody and detected by Western blotting with anti-V5 and anti-FLAG antibodies. N = 4. **(E)** Effect of MLSA-1 on Lamtor1-G2A. The chemotaxis in response to CCL2 (25 ng/ml) of Lamtor1-KO-G2A-THP1 cells was assessed under specific concentrations of MLSA-1, using the Transwell assay system (pore size of 5 $\mu$m). N = 5. **(F)** Effect of MLSA-1 on MLC phosphorylation of Lamtor1-G2A. Lamtor1-KO-G2A-THP1 cells were treated with an indicated concentration of MLSA-1 for 6 h, and MLC phosphorylation was evaluated by Western blotting using an anti-PMLC antibody. N = 4. **(A, B, C, D, E, F)** Data information: Statistical analyses were performed using the two-sided Mann–Whitney $U$ test (A) (median; 25th and 75th percentiles; and minimum and maximum of a population excluding outliers) and a two-sided $t$ test (B, C, D, E, F) (means ± s.d).
Source data are available for this figure.

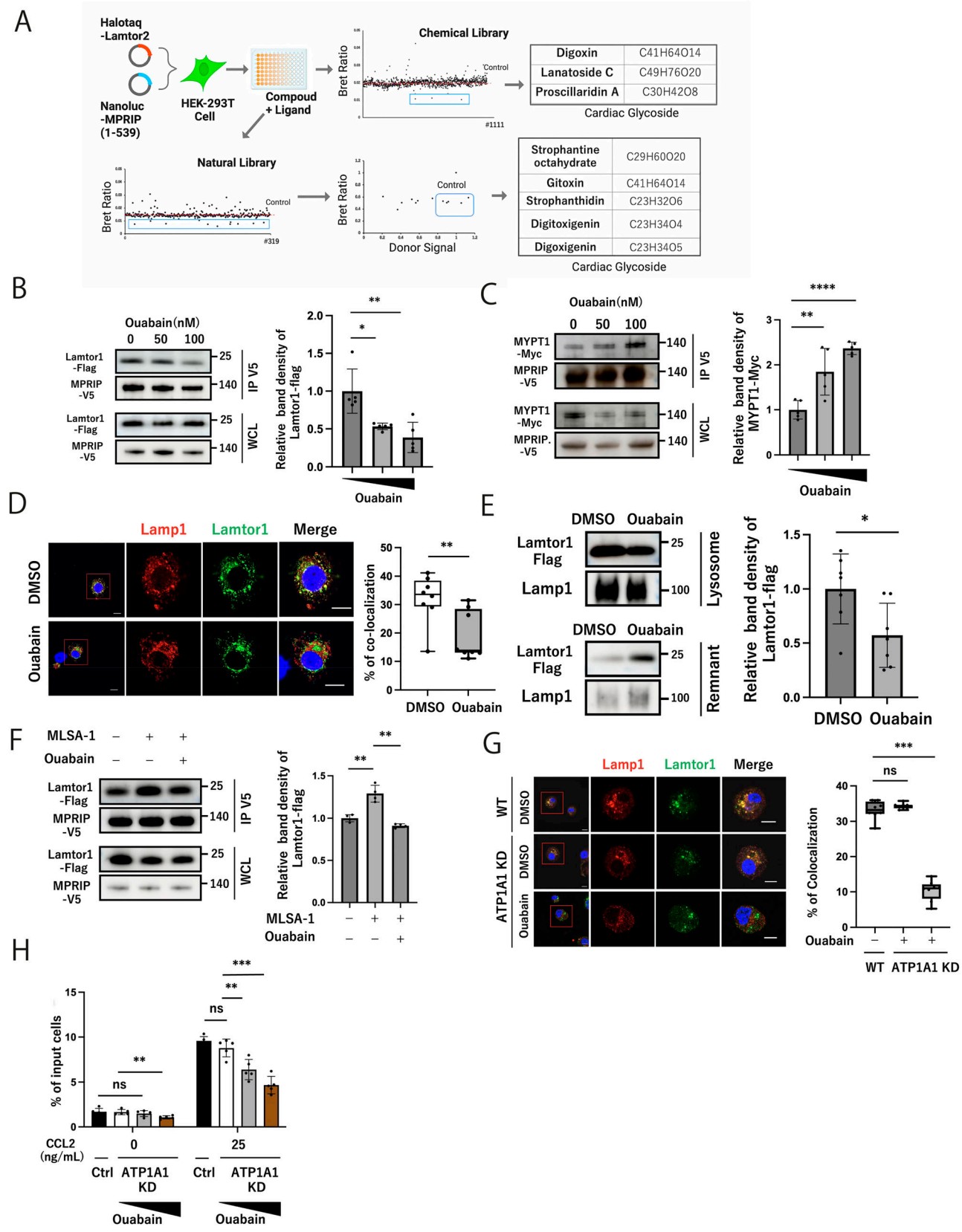

ouabain (Fig S4B). Consistently, MLC phosphorylation did not differ regardless of the presence of ouabain in Lamtor1-KO THP1 or MPRIP-KO THP1 cells (Fig S4C). These results suggest that ouabain suppressed cell migration dependent on Lamtor1 and MPRIP.

We then investigated the therapeutic potential of ouabain in modulating the immune response. First, CFSE-labeled dendritic cells (DCs) with or without ouabain treatment were injected into the footpad of mice, and the migrated DCs in the draining lymph nodes were counted. The migration of ouabain-treated DCs to the draining lymph nodes was impaired (Fig 4D). We further investigated the effects of ouabain in three different acute inflammation models mediated by leukocyte infiltration: injection of monosodium urate crystals into the ankle (a gouty arthritis model); injection of LPS into the peritoneal cavity followed by evaluation of the lung (an acute lung injury model); and injection of aluminum hydroxide (alum) into the peritoneal cavity to promote neutrophil infiltration. In the gouty arthritis model, ouabain-treated mice showed a marked decrease in joint swelling and cellular infiltration (Fig 4E). In the LPS-induced lung injury model, LPS administration induced severe lung inflammation with alveolar cell stromal thickening and leukocyte infiltration; however, ouabain ameliorated lung inflammation and reduced leukocyte infiltration (Fig 4F). In the alum-induced peritonitis model, ouabain-treated mice exhibited lower neutrophil infiltration (Fig 4G). However, ouabain did not influence the production of cytokines such as TNF-α (Fig S4D) and the expression levels of CD80/CD86 in 6- and 24-h LPS-treated DCs (Fig S4E). These results suggest that ouabain suppresses immune cell recruitment rather than leukocyte activation. Hence, ouabain is a potential treatment for inflammatory diseases by suppressing leukocyte recruitment.

## Discussion

In this study, we investigated the role of calcium signaling in the regulation of actomyosin-mediated cell migration, particularly through the interaction between the Ragulator complex and MPRIP. We found that (1) lysosomal calcium efflux through TRPML1 can enhance the interaction between the Ragulator complex and MPRIP, thereby facilitating cell motility; (2) migration of the Ragulator complex from the lysosome diminishes these effects; (3) ouabain, a cardiac glycoside, diminishes the interaction of the Ragulator complex with MPRIP by preventing the localization of the Ragulator complex to lysosomes; and (4) ouabain alleviates in vivo inflammation by suppressing immune cell recruitment. Lysosomes provide Ca2+ through TRPML1, which allows the interaction of the Ragulator complex with MPRIP to promote actomyosin-mediated cell migration, whereas ouabain alleviates inflammation by disrupting these equilibria.

Although the ER is considered the primary intracellular source of cellular calcium (Berridge, 2009), lysosomal calcium efflux is thought to act as a second messenger for local signal transduction. Because the concentration of Ca2+ from the calcium channels falls steeply with distance because of cytoplasmic Ca2+ buffering (Neher, 1998), the effector molecule must stay close to the Ca2+ micro-domain brought about by Ca2+ channels. We demonstrated that Lamtor1 colocalizes and interacts with TRPML1, and the modulation of calcium efflux through TRPML1 enhances the interaction between the Ragulator complex and MPRIP. This effect was more pronounced than that of A23187, which facilitates calcium transport across the entire cell membrane. Consistently, the proximity of Lamtor1 to TRPML1 has been reported to regulate lysosomal trafficking (Sun et al, 2022). Calcium signaling has also been implicated in cell motility (Vicente-Manzanares et al, 2009). TRPML1 regulates membrane trafficking, signal transduction, and ion homeostasis (Dong et al, 2010). In addition, it activates myosin IIA and stabilizes F-actin filaments at the rear of DCs. During cell migration, proximal localization of myosin IIA and lysosomes was observed at the rear of DCs in WT cells; however, in the absence of TRPML1, myosin IIA hardly colocalized with lysosomes, despite the normal localization of lysosomes (Bretou et al, 2017). This suggests a specific molecular mechanism wherein lysosomes act as an activation platform for myosin IIA regulated by the TRPML1 channel. MPRIP binds to the actin bundles, bridging actin and myosin to interact with MYPT1 and anchoring MLCP to keep myosin IIA inactive. During cell migration, the Ragulator complex interacts with MPRIP, interfering with the MPRIP-MYPT1 interaction, thereby sequestering MLCP from the actin bundle and resulting in myosin IIA activation. In this study, we

---

**Figure 3. Ouabain inhibits the Lamtor1–MPRIP interaction by preventing the localization of Lamtor1 to lysosomes.**
**(A)** Screening schematic for identifying inhibitors of the Ragulator complex and MPRIP interaction using a library of chemical and phytochemical compounds by NanoBRET between Lamtor2 and MPRIP1-539. After excluding compounds that reduced the Nanoluciferase signal, candidate compounds are listed. **(B)** Effect of ouabain on the interaction between Lamtor1 and MPRIP. The V5-tagged MPRIP expression vector was transfected into Lamtor1-Flag-expressed HEK293T cells. After 6 h of treatment with ouabain (50 or 100 nM), cells were lysed, immunoprecipitated with an anti-V5 antibody, and detected by Western blotting with anti-V5 and anti-FLAG antibodies. N = 5. **(C)** Effect of ouabain on the MYPT1–MPRIP interaction. The Myc-tagged MYPT1 expression vector was transfected into V5-tagged MPRIP-expressed HEK293T cells. After 6 h of treatment with ouabain (50 or 100 nM), the cells were lysed, immunoprecipitated with an anti-V5 antibody, and detected by Western blotting with anti-V5 and anti-Myc antibodies. N = 5. **(D)** Effect of ouabain on Lamtor1 localization in lysosomes. WT-THP1 cells were pretreated with 100 nM ouabain or DMSO for 6 h. Localization of Lamtor1 and Lamp1 was assessed by confocal microscopy using anti-Lamtor1 (green) and anti-Lamp1 (red) antibodies. Representative images are shown. Scale bar, 10 μm. N = 8. **(E)** Localization of Lamtor1 after ouabain treatment. Lysosomes of Lamtor1-KO-Full-THP1 cells were isolated, and the expression of Lamtor1-Flag in the lysosomal (upper) and remnant (down) fractions was evaluated by Western blotting using anti-FLAG and anti-Lamp1 antibodies. N = 7. **(F)** Attenuation of MLSA-1–mediated Lamtor1–MPRIP interaction by ouabain. HEK293T cells expressing Lamtor1-Flag were transfected with MPRIP-V5 and pretreated with MLSA-1 50 μM and with or without ouabain 100 nM for 6 h. The cells were lysed, immunoprecipitated with an anti-V5 antibody, and detected by Western blotting using anti-V5 and anti-FLAG antibodies. N = 4. **(G)** Effect of ouabain on Lamtor1 localization to lysosomes after ATP1A1 receptor knockdown. ATP1A1-knockdown (ATP1A1-KD)-THP1 cells were pretreated with ouabain 100 nM or DMSO for 6 h. Localization of Lamtor1 to the lysosomes was evaluated by confocal microscopy using anti-Lamtor1 (green) and anti-Lamp1 (red) antibodies. Scale bar, 10 μm. N = 8. **(H)** Effects of ATP1A1 receptor knockdown on cell migration. ATP1A1-knockdown (ATP1A1-KD)-THP1 cells were pretreated with ouabain (50 or 100 nM) or DMSO for 2 h. Chemotaxis was induced using 25 ng/ml MCP-1. N = 5. **(A, B, C, D, E, F, G, H)** Data information: Statistical analyses were performed by a two-sided t test (A, B, C, E, F, H) (means ± s.d) or a two-sided Mann–Whitney U test (D, G) (median; 25th and 75th percentiles; and minimum and maximum of a population excluding outliers).
Source data are available for this figure.

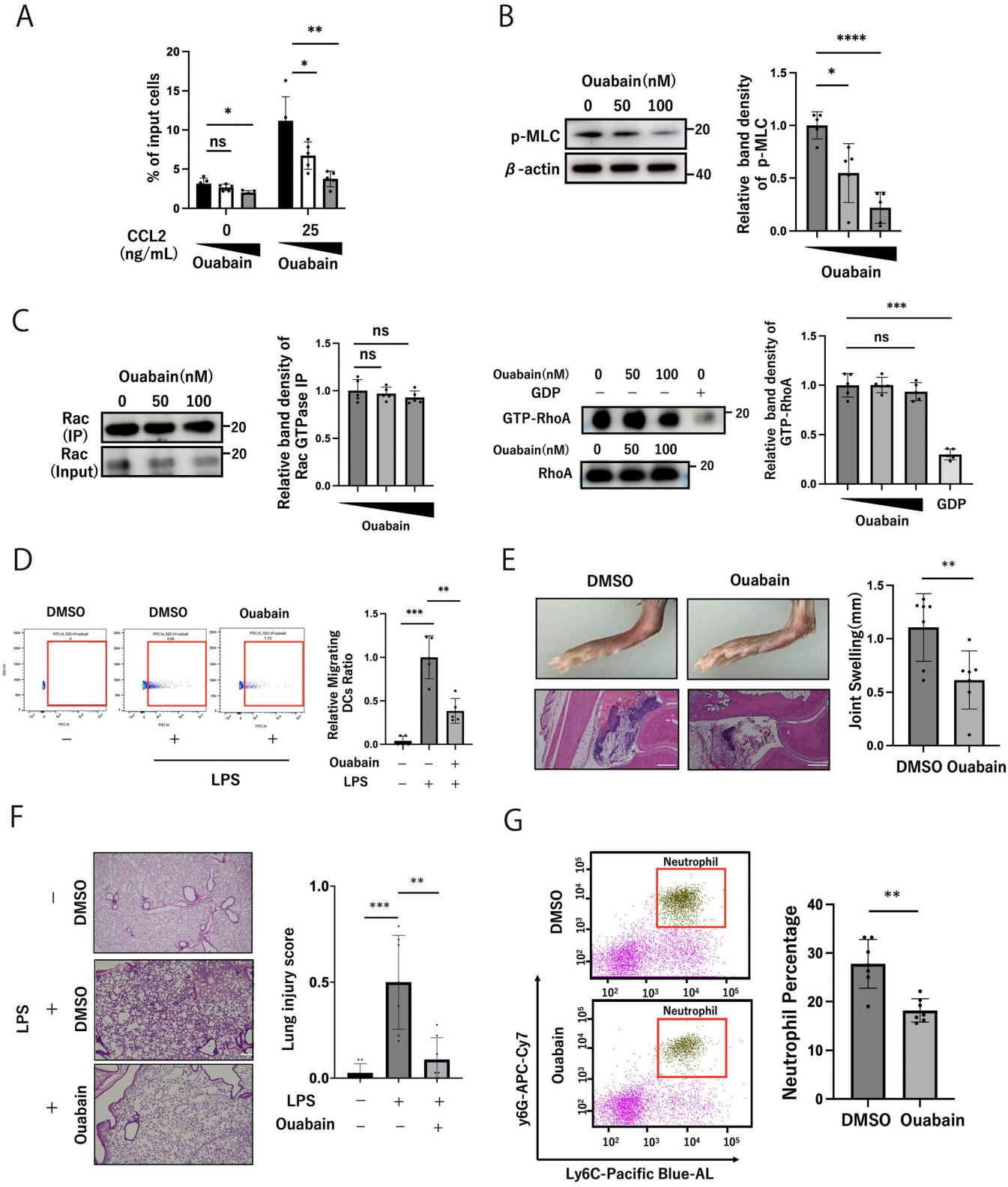

**Figure 4. Ouabain ameliorates inflammation by inhibiting cell migration.**
**(A)** Effect of ouabain on migration of WT-THP1 cells. WT-THP1 cells were preincubated with ouabain (50 or 100 nM) or DMSO for 2 h. Chemotaxis was induced in response to CCL2 (25 ng/ml). N = 5. **(B)** Effects of ouabain on MLC phosphorylation. WT-THP1 cells treated with ouabain (50 or 100 nM) or DMSO for 6 h were lysed, and Western blotting was performed using an anti-PMLC antibody. N = 5. **(C)** Detection of active Rac1 and RhoA. Rac1 expression was assessed using Active Rac1 Detection Kit. Representative Western blots are shown, using the anti-Rac1 antibodies at left. N = 5. Active RhoA levels were assessed using Active RhoA Detection Kit. Representative

showed that increased Ca2+ efflux via TRPML1 enhanced the interaction between MPRIP and the Ragulator complex and promoted cell migration. Furthermore, defective lysosomal localization of the Ragulator complex canceled the TRPML1-mediated activation of myosin IIA and the Ragulator complex–MPRIP interaction. Our findings suggest an alternative mechanism for myosin IIA activation via lysosomal TRPML1. Thus, lysosomal Ca2+ efflux via TRPML1 promotes the interaction between the Ragulator complex on the lysosomal membrane and MPRIP bound to the actin bundle, resulting in the dissociation of MLCP from MPRIP, which in turn activates myosin IIA and actomyosin contraction.

For clinical applications targeting cell migration, treatments that focus on the integrins $\alpha IIb\beta 3$, $\alpha 4\beta 7/\alpha 4\beta 1$, and $\alpha L\beta 2$ have been successfully marketed (Slack et al, 2022). However, its use is limited by their high costs and rare but serious complications such as progressive multifocal leukoencephalopathy (Kartau et al, 2019; Vivekanandan et al, 2021). We identified the cardiac glycoside ouabain as an inhibitor of the Ragulator complex–MPRIP interaction, providing new insights into therapeutic strategies targeting integrin-independent cell migration. Cardiac glycosides have traditionally been used as inhibitors of Na+/K+-ATPase, which increases cellular and sarcoplasmic reticulum (SR) calcium levels. Increased calcium binds to troponin C to decrease actin–myosin cross-linking, resulting in inotropic effects by enhancing the actomyosin contractile force. In this study, we showed that ouabain inhibits myosin IIA activity in leukocytes. The role of ouabain in actomyosin contraction in cardiomyocytes and leukocytes seems to be the opposite. Cardiac glycosides inhibit Ca2+ efflux in cardiomyocytes by inhibiting Na+/K+-ATPase at relatively high concentrations; in our study, ouabain prevents the localization of the Ragulator complex to lysosomes independent of the ATP1A1 channel in leukocytes, diminishing its interaction with MPRIP; this suggests that the effect of ouabain may differ depending on the cell type. Cardiac glycoside has also been reported to suppress cell migration and invasion of lung cancer cells by inhibiting the ERK signaling pathway (Pongrakhananon et al, 2013), as well as reduce inflammation in a rheumatoid arthritis model by inhibiting the NF-κB signaling pathway (Menger et al, 2012). In our study, the concentration of ouabain was relatively low compared with that in cardiomyocytes, and we evaluated the earlier response of immune cells to ouabain treatment compared with other studies. After all, it is probable that cardiac glycosides cause different effects depending on the experimental setting; however, in any case, they could suppress cell motility and inflammation. Regarding the therapeutic action and safety of ouabain, the mice in our study were

treated with ouabain for 24 or 36 h; thus, we believed that the action of ouabain was primarily mediated via cellular dynamics. However, there is a concern about the long-term effects of ouabain because the short-term treatment had no effect on the phosphorylation of ppS6K, but in the 24-h treatment, it gradually decreased. Considering its application for treating human diseases, the evaluation of long-term effects of ouabain is needed.

It has been shown that the Ragulator complex, composed of five subunits, forms a scaffold on the lysosomal membrane. Thus, there may be several possibilities other than proximity to Ca2+ channels as to why the loss of Lamtor1 localization to lysosomes inhibits actomyosin contraction: (1) the impaired localization of Lamtor1 to lysosomes may disrupt the formation of the Ragulator complex; (2) the loss of localization to lysosomes may reduce binding affinity to MPRIP because of the aberrant formation of the Ragulator complex; and (3) mTORC1 may be involved in the actomyosin contraction. We have confirmed that Lamtor1 associated with Lamtor2-5 even in the Lamtor1-G2A mutant, suggesting the Ragulator complex formation in the Lamtor1-G2A mutant cells. In the immunoprecipitation assay, we observed the shift of Lamtor1-G2A binding size, suggesting that Lamtor1-G2A may aggregate other proteins and reduce the binding affinity to MPRIP. In addition, we observed that ouabain did not alter the mTORC1 activity in the short term and we have previously reported a comparable actomyosin contraction, regardless of treatment with rapamycin and Torin1 (Nakatani et al, 2021), suggesting that mTORC1 is not involved in the cell motility. Regarding the pharmacological action of ouabain in terms of lysosomal localization of Lamtor1, because palmitoylation of Lamtor1 has been reported to be crucial for stabilization of the Ragulator complex on lysosomal membranes (Nada et al, 2009), ouabain appears to regulate the palmitoylation of Lamtor1, affecting the presence of Lamtor1 in lysosomes, although this has not yet been addressed. Further research is needed to unravel the precise molecular mechanisms by which ouabain regulates lysosomal localization of the Ragulator complex.

Collectively, we revealed that the interaction between the lysosomal Ragulator complex and MPRIP connects two crucial factors to promote cell migration: lysosomal Ca2+ efflux and myosin IIA–mediated actomyosin contraction. Furthermore, ouabain inhibited cell migration by dissociating the Ragulator complex from the lysosomal membrane. These findings not only advance our understanding of the fundamental processes governing cell motility but also pave the way for developing novel therapeutic strategies targeting aberrant cell migration.

---

Western blots are shown, using the anti-RhoA antibodies at right. N = 5. **(D)** Effect of ouabain on DC migration in LNs. CFSE-labeled WT DCs were pretreated with 200 nM ouabain or DMSO and adoptively injected into the footpads of WT recipient mice. After 48 h, cells from popliteal LNs were isolated and CFSE+ cells were counted. The ratio of ouabain-treated DCs to DMSO-treated DCs is also shown. N = 5. **(E)** Effect of ouabain in an acute gouty arthritis model. MSU crystals suspended in endotoxin-free PBS, with or without ouabain (10 $\mu M$ and 0.2 $\mu l$), were injected into the ankles of C57BL/6 WT mice. Ankle joint swelling at 24 h was measured using electronic calipers, and ankle joints were stained with hematoxylin and eosin. N = 7. **(F)** Effect of ouabain in the LPS-induced acute lung injury model. LPS was administered with or without ouabain (1 $\mu g/kg$) to the peritoneum of C57BL/6 WT mice. After 36 h, lung tissues were collected, and photomicrographs of tissues stained with hematoxylin and eosin were taken. The lung injury scores were calculated and compared between the ouabain and DMSO groups. N = 7. **(G)** Effects of ouabain on neutrophil recruitment in an Alum-induced peritonitis model. 12 h after the peritoneal injection of 1 $\mu g/kg$ ouabain, 600 $\mu l$ alum solution (20 mg/ml) was injected intraperitoneally. After 4 h, the peritoneal fluid was isolated, and the percentage of CD11b+ Ly6G+ neutrophils in the infiltrating cells was measured by FACS. N = 7. **(A, B, C, D, E, F, G)** Data information: Statistical analyses were performed by a two-sided $t$ test (A, B, C, D) (means ± s.d) or a two-sided Mann–Whitney $U$ test (E, F, G) (median; 25th and 75th percentiles; and minimum and maximum of a population excluding outliers).

Source data are available for this figure.

# Materials and Methods

### Mice

C57BL/6J mice were used for in vivo experiments. Mice were housed under specific pathogen-free conditions with a 12-h light/dark cycle, at a temperature of 22 ± 2°C, and a relative humidity of 50 ± 5%. The mice were fed a standard mouse chow diet, and three to five mice were housed in the same cage. 8–9-wk-old mice were used in the experiments. Both male and female mice were used; however, the sex was always matched in each experiment. Animal experiments were approved by the ethics board of the Graduate School of Medicine, Osaka University (28-008-034), and all animal experiments were performed according to Osaka University's regulations.

### Cells and cell culture

THP1 cells, a human acute monocyte leukemia cell line, were obtained from the ATCC (TIB-202) and cultured in RPMI 1640 (Nacalai Tesque) supplemented with 10% FBS, 1% penicillin/streptomycin, and 0.05 mM 2-mercaptoethanol. THP1 cells were differentiated with 50 nM phorbol 12-myristate 13-acetate (PMA; Sigma-Aldrich) for 72 h. HEK293T cells were obtained from the ATCC (CRL-3216) and cultured in DMEM supplemented with 10% FCS and 1% penicillin and streptomycin. All cells were maintained at 37°C in 5% $CO_2$. BMDCs were generated by culturing bone marrow cells with 50 $\mu$g/ml GM-CSF (R&D Systems) for 6–8 d as described previously (Inaba et al, 1992).

### Antibodies, reagents, and fluorescent dyes

Reagents were obtained from the suppliers listed in Table S1.

### NanoBRET

Plasmids of MPRIP1-539 tagged with a Halo-Tag donor at the amino terminus (MPRIP1-539-HTC), Lamtor1 tagged with a NanoLuc acceptor at the amino terminus (Lamtor1-NLF-C), and Lamtor2 tagged with a NanoLuc acceptor at the amino terminus (Lamtor2-NLF-C) were generated using In-Fusion HD Cloning Kit (Takara Bio). Briefly, cDNA fragments of human Lamtor1, Lamtor2, and MPRIP1-539 were ligated to homologous sequences by overlapping PCR using the primers listed in Table S1. These fragments were then cloned into the EcoRI-SacI sites of the pNLF1-C [CMV/Hygro] vector (Accession Number KF811458) for Lamtor1 and Lamtor2 and into the EcoRI-SacI sites of the pHTC Halo-Tag CMV-neo-Vector (Accession Number JF920305) for MPRIP1-539, using In-Fusion HD Cloning Kit. Transfection of HEK293T cells for luciferase assays was performed using Lipofectamine 2000 (Invitrogen). After 48 h of transfection, luminescence was measured after the addition of Nano-Glo Luciferase Assay Substrate (Promega) using GloMax Discover Microplate Reader (Promega). For evaluating the effect of TRPML1, HEK293T cells transiently expressing MPRIP1-539-HTC and Lamtor1-NLF-C were treated overnight with MLSA-1 8 $\mu$M and MLSI-3 16 $\mu$M before the assay.

### Chemical compound library screening

A phytochemical and chemical library was obtained from Prestwick Chemical. HEK293T cells transiently expressing MPRIP1-539-HTC and Lamtor2-NLF-C were treated overnight with chemical compounds diluted to a concentration of 10 $\mu$M with DMEM. MPRIP1-539–Lamtor1 interactions were examined using the NanoBRET assay, as described above. The result is listed as Supplemental Data 1 and Supplemental Data 2.

### Western blotting

Cells were solubilized in buffer A (1% Nonidet P-40 [NP-40], 50 mM Tris–HCl [pH 7.4], 150 mM NaCl, 1 mM EDTA, 5% glycerol, and 2% n-octyl-b-D-glucopyranoside) with proteinase inhibitor (Roche), vortexed for 10 min, and centrifuged at 20,000$g$ for 15 min at 4°C. Supernatants were mixed with 2× Laemmli sample buffer containing 2-mercaptoethanol and denatured for 5 min at 95°C. The reduced samples were electrophoresed on 4–12% Bis-Tris gels (Life Technologies), transferred to nitrocellulose membranes, and blotted with the antibodies listed in Table S1. Protein concentrations in the SDS–PAGE gel bands were determined using ImageJ software (NIH).

### Immunoprecipitation

HEK293T cells were transfected with the indicated plasmids using Lipofectamine 2000 (Invitrogen) and incubated for 48 h. Cells were solubilized with buffer A and centrifuged at 20,000$g$ for 15 min at 4°C. Immunoprecipitation was performed using Dynabeads Protein G Immunoprecipitation Kit (10007D; Invitrogen). Briefly, after binding of the antibody to magnetic beads (15 min at room temperature, with rotation), cell lysates and antibody-coated beads were mixed and incubated for 1 h at room temperature. For the pharmacological assay, HEK293T cells were treated with ouabain (0, 50, and 100 nM) and MLSA-1(0, 20, and 50 $\mu$M) at specified concentrations for 6 h before the immunoprecipitation assay. For the $Ca^{2+}$ assay, 3 mM EDTA was added to the cell lysate and incubated for 6 h, followed by the addition of $CaCl_2$ (0, 50, and 100 $\mu$M) to the lysate at 4°C for 12 h. For the Ragulator complex formation assay, HEK293T cells were transfected with Lamtor1-G2A-Flag with DMSO pretreatment for 6 h, Lamtor1-Full-Flag with DMSO pretreatment for 6 h, or Lamtor1-Full-Flag with ouabain 100 nM pretreatment for 6 h. Immunoprecipitation was performed with DYKDDDK-coated beads overnight at 4°C.

All immunoprecipitated samples were washed three times, and proteins were eluted with 2× Laemmli sample buffer containing 2-mercaptoethanol and denatured for 5 min at 95°C. The samples were separated using sodium dodecyl sulfate–polyacrylamide gel electrophoresis and blotted with the indicated antibodies.

### Lysosomal fractionation

THP1 cells were cultivated at a density of 1.0 × 10$^6$ cells/ml, and lysosomes were isolated from 3 × 10$^6$ cultured cells by density gradient separation (14,5000$g$, 2 h), using Lysosome Enrichment Kit for Tissue and Cultured Cells (Pierce Biotechnology) according to

the manufacturer's protocol. The lysosome fraction band in the top 2 ml of the gradient was centrifuged as a lysosome pellet, and the remaining fraction band was centrifuged as a remnant pellet. Both pellets were mixed with 2× Laemmli sample buffer containing 2-mercaptoethanol and denatured for 5 min at 95°C. Samples were separated using sodium dodecyl sulfate–polyacrylamide gel electrophoresis and blotted with the indicated antibodies. For the pharmacological assay, before the assay, THP1 cells were treated with 100 nM ouabain for 6 h.

### In vitro cell migration assay

For the Transwell assay, inserts (pore size, 5.0 $\mu$m for THP1; Corning) were placed in 24-well plates filled with 0.6 ml of 0.1% (wt/vol) BSA in RPMI medium containing recombinant CCL2. A cell suspension of THP1 ($2 \times 10^5$ cells/100 $\mu$l) was added to the upper well, followed by incubation for 4 h at 37°C. The cells in the lower chamber were detached for 5 min using 5 mM PBS-EDTA and counted using the Guava PCA system. Migration ability (% of input cells) was evaluated by dividing the number of cells in the lower chamber by the number of cells in the upper chamber and multiplying it by 100. For the pharmacological assay, THP1 cells were treated with ouabain (0, 50, and 100 nM) for 2 h before the assay, and MLSI-3 (0, 8, and 16 $\mu$M) and MLSA-1 (0, 4, and 8 $\mu$M) were added at the start of the assay. These substances were present in both the upper and lower chambers during the assay. For DC migration in 3D collagen matrices, chemotaxis assays were performed using $\mu$-Slide Chemotaxis (ibidi GmbH) according to the manufacturer's protocol. Briefly, LPS (200 ng/ml)-treated BMDCs were suspended in 1.5 mg/ml type I collagen (BD Biosciences) containing 5% FCS and seeded to the observation channel, incubating for 10 min. Subsequently, RPMI medium containing 0.1% BSA was filled to both sides of the chamber, with one side containing CCL19 (5 $\mu$g/ml). After 30 min of incubation, DC locomotion was examined at 1-min intervals by confocal time-lapse video microscopy. The velocity of cells was measured using ImageJ. To visualize lysosomes, BMDCs were labeled with 100 nM LysoTracker Red DND-99 for 30 min before being subjected to chemotaxis assays in $\mu$-Slide Chemotaxis. For the pharmacological assay, MLSA (8 $\mu$M), MLSI-3 (16 $\mu$M), and blebbistatin (50 $\mu$M) were treated with BMDC for 2 h before and during the assay.

### Expression vector cloning

cDNAs encoding Lamtor1 (Met1–Pro161) and its G2A mutant variant, both with a C-terminal 3×FLAG tag, have been previously described (Nakatani et al, 2021; Tsujimoto et al, 2023). Lamtor2–5, each tagged at the C terminus with 3×FLAG, were synthesized by Eurofins Genomics and cloned into the NotI–BamHI sites of the CSII-EF-IRES2-Venus vector (provided by Dr. Hiroyuki Miyoshi, Keio University) using In-Fusion HD Cloning Kit. The previously characterized V5-tagged MPRIP (Met1–Asp2457)–IRES-RFP lentiviral vector (Nakatani et al, 2021) was subcloned into the same sites as the CSII-EF-IRES2-Venus vector. cDNAs encoding the truncated forms of MPRIP (MPRIP$^{1-150}$, MPRIP$^{1-384}$, MPRIP$^{1-545}$, and MPRIP$^{545-1024}$) with the V5 sequence at the C terminus were amplified by overlapping PCR using the completed MPRIP-V5 vector as a template and

subsequently cloned into the same vector sites. The cloned primers are listed in Table S1. TRPML1 tagged at the C terminus with V5 was synthesized by Eurofins Genomics and cloned into the NotI–BamHI sites of the CSII-EF-IRES2-Venus vector. The MYPT1-Myc vector (VB190904-1066bwp, pLV[Exp]-EGFP-EF1A> hPPP1R12A [NM_002480.3]/Myc) was purchased commercially and cloned. These vectors and constructs were used to express the respective proteins in specific cell lines for subsequent functional assays.

### Generation of KO or knockdown cell lines

Lamtor1-deficient THP1 cells were generated as described previously (Nakatani et al, 2021). MPRIP-deficient cell lines were generated by transfecting cells stably expressing Cas9 with a mixture of several gRNAs using the Avalanche-Omni transfection reagent (EZ Biosystems). The target sequences used for MPRIP-KO are listed in Table S1. ATP1A1 knockdown was performed using lentiviral plasmids expressing shRNAs specific for human ATP1A1; the target sequence is listed in Table S1 (TRCN0000043226; Sigma-Aldrich).

### Establishment of stable transfectants

Lamtor1−/− THP1 transfectants stably expressing full-length or mutant Lamtor1 were established by lentivirally introducing FLAG-tagged full-length Lamtor1 (Lamtor1-KO-Full-THP1) or FLAG-tagged G2A-Lamtor1 (Lamtor1-KO-G2A-THP1) into Lamtor1−/− THP1 cells, and single clones were screened by Western blotting with anti-FLAG mAb (M2; Sigma-Aldrich).

### Enzyme-linked immunosorbent assay

Serum cytokine levels (TNF-$\alpha$) were measured with DuoSet ELISA Development System (R&D Systems) following the product manual.

### Immunostaining and fluorescence microscopy

For immunostaining, cells were fixed with 4% paraformaldehyde and permeabilized with 0.1% Triton X-100 in PBS before incubation with primary antibodies, followed by incubation with Alexa Fluor secondary antibodies. Nuclei were stained with DAPI in a mounting medium (ProLong Gold with DAPI; Thermo Fisher Scientific). For imaging with multiple channels, extensive control experiments were performed to ensure that there was no nonspecific staining or crosstalk between channels. These control experiments included the following: (a) cells that lacked one of the proteins of interest and (b) staining without one of the primary or secondary antibodies. Fluorescence images of the fixed cells were obtained using an FV3000 confocal laser scanning microscope (Olympus).

### Pharmacological application in cells

HEK293T cells were treated with ouabain, MLSI-3, or MLSA-1 at specified concentrations for 6 h. Immunoprecipitation and lysosomal enrichment assays were then performed. THP1 cells were pretreated with ouabain for 6 h, and the targets included phosphorylation levels of MLC and Rac GTPase, cell cytotoxicity, TNF-$\alpha$ activity, and localization of the indicated targets, which were

assayed as mentioned above. Regarding the migration assay, the indicated concentrations and durations have been explained in their respective sections.

## Active Rac1 detection

WT-THP1 cells were pretreated with 100 nM ouabain or DMSO for 6 h and lysed. The active form of Rac1 was pulled down with GST-PAK1-PBD fusion protein and immunoprecipitated with glutathione resin (Active Rac1 Detection Kit; Cell Signaling Technologies). Active Rac1 levels were determined by Western blotting using a Rac1/Cdc42 antibody (Cell Signaling Technologies).

## RhoA pull-down assay

WT-THP1 cells were pretreated with 100 nM of ouabain or DMSO for 6 h and lysed. Then, the active form of RhoA was pulled down with a GST-Rhotekin-RBD fusion protein and immunoprecipitated with glutathione resin (Active RhoA Detection Kit; Cell Signaling Technologies). Active RhoA levels were determined by Western blotting using an anti-rabbit RhoA antibody (Cell Signaling Technologies). As a negative control, WT-THP1 cells were loaded with GDP before pull-down.

## Adhesion assay

For the THP1 cell adhesion assay, 96-well plates were coated with VCAM (10 $\mu$g/ml). THP1 cells were stained with calcein-AM (5 mM) and suspended in RPMI medium supplemented with 0.1% BSA at a concentration of $5 \times 10^5$ cells/well. The cell suspension was added to VCAM-coated wells and simultaneously treated with either MLSI-3 (16 $\mu$M) or MLSA-1 (8 $\mu$M). After incubation at 37°C for 2 h, non-adherent cells were removed by washing with PBS, and adherent cells were lysed in 0.1% NP-40. For the total group, $5 \times 10^5$ THP1 cells were added to uncoated wells and lysed in 0.1% NP-40. Fluorescence intensity of each well was measured using a microplate reader set to an excitation wavelength of 485 nm. The adhesion rate was calculated by dividing the fluorescence intensity of each sample by that of the total group.

## Fluorescein-labeled DC-trafficking assay

For the fluorescein-labeled DC-trafficking assay, $3 \times 10^6$ BMDCs were pretreated with ouabain 200 nM for 6 h and subsequently treated with LPS 200 ng/ml for 2 h. Next, CFSE was labeled with 5 $\mu$M CFSE and injected into the footpads of WT mice. After 48 h, popliteal LNs were collected, and the frequency of CFSE-positive cells was assessed by flow cytometry.

## Crystal-induced ankle arthritis model

C57BL/6 WT mice were subcutaneously anesthetized with ketamine and xylazine. Monosodium urate crystals (0.5 mg) suspended in 20 $\mu$l endotoxin-free PBS were co-administered with ouabain (10 $\mu$M, 0.2 $\mu$l) or DMSO (0.2 $\mu$l), and injected intra-articularly into the right tibia–tarsal joints. After 24 h, the joint sizes were measured

using an electronic caliper, and the mice were euthanized to collect the joints for hematoxylin and eosin staining.

## LPS-induced lung injury model

C57BL/6 WT mice were intraperitoneally injected with LPS (10 mg/kg). To assess the effect of ouabain on sepsis-related lung infiltration, 1 $\mu$g/kg of ouabain was co-administered intraperitoneally with LPS. At 36 h post-administration, the mice were euthanized, and lung tissues were fixed by intratracheal instillation of 4% paraformaldehyde. The tissues were stained with hematoxylin and eosin (HE) and examined by optical microscopy. Lung injury scores were quantified by a staff member who was blinded to the treatment groups, following the guidelines provided by the American Thoracic Society (Matute-Bello et al, 2011).

## Mouse peritonitis model

C57BL/6 WT mice were intraperitoneally injected with 600 $\mu$l alum solution (20 mg/ml). Then, 1 $\mu$g/kg of ouabain was co-administered intraperitoneally with alum. To screen for immune cell populations, peritoneal infiltrating cells were stained with the indicated antibodies and analyzed using a FACSCanto II (BD Biosciences).

## Flow cytometry

To detect the expression of CD80/CD86 in BMDCs, the cells were first stimulated with LPS for 6 and 24 h and simultaneously treated with 100 nM ouabain or DMSO for 6 h, then stained with the indicated antibodies. For the DC-trafficking assay, CFSE-positive cells were assessed by flow cytometry. Data from both experiments were collected by a FACSCanto II (BD Biosciences) by FACSDiva (v5.0.3) software and analyzed by FlowJo (v10) software.

## RNA-seq

Overall $2 \times 10^6$ Lamtor1-KO-Full and Lamtor1-KO-G2A-THP1 cells were collected for RNA extraction. Sequencing and data analysis were performed by the Genome Information Research Center at the Research Institute for Microbial Diseases, Osaka University. Quality control of the raw reads was performed using Trimmomatic (v0.39), and that of the resulting clean reads was performed with HISAT2 (v2.1.0). Gene expression analysis was performed by featureCounts (v2.0.6) and Cuffdiff (v2.2.1). All analysis data were evaluated and visualized by the server of RNAseqChef (Etoh & Nakao, 2023).

## Statistical analysis

All statistical analyses were performed using GraphPad Prism version 8. Normally distributed data were compared using a *t* test and are presented as means ± SEM. For non-normally distributed data, the Mann–Whitney *U* test was performed. Results for non-normal distributions are presented as the median with 25th and 75th percentiles, and minimum and maximum values, excluding outliers. Statistical significance was set at $P < 0.05$ for all analyses.

# Data Availability

All data that support the conclusions are available from the correspondence author on reasonable request.

# Supplementary Information

# Acknowledgements

We would like to thank Shigeyuki Nada for supporting our project. This study was supported in part by research grants from Japan Society for the Promotion of Science (JSPS) KAKENHI (18K08386 and 22H03111 to H Takamatsu, JP18H05282 to A Kumanogoh), JSPS Core-to-Core Program (JPJSCCA20210008 to A Kumanogoh), a Core Research for Evolutionary Science and Technology (CREST) grant from JST (JPMJCR16G2 to H Takamatsu), the Japan Agency for Medical Research and Development (AMED)-CREST (15652237, 22gm1810003h0001 to A Kumanogoh), AMED (23gm4010022h0001 to H Takamatsu, J200705023, J200705710, J200705049, JP18cm016335, JP18cm059042, JP223fa627002, and 24ek0410124h0001 to A Kumanogoh), and the Center of Innovation Program (COI-STREAM) from the Ministry of Education, Culture, Sports, Science and Technology of Japan (MEXT) (to A Kumanogoh). We thank Editage (www.editage.jp) for English-language editing.

## Author Contributions

T Jo: conceptualization, resources, formal analysis, validation, investigation, visualization, methodology, and writing—original draft, review, and editing.
K Tsujimoto: methodology.
T Nakatani: methodology.
D Nagira: methodology.
Y Muto: methodology.
T Hirayama: methodology.
H Konaka: methodology.
M Okada: methodology.
H Takamatsu: conceptualization, supervision, funding acquisition, investigation, methodology, project administration, and writing—review and editing.
A Kumanogoh: conceptualization, supervision, funding acquisition, and project administration.

## Conflict of Interest Statement

The authors declare that they have no conflict of interest.

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
