## [Reviewer comments · Life Science Alliance]

Life Science Alliance

The Ragulator complex and lysosomal calcium release are crucial for cell migration.

Tatsunori Jo, Kohei Tsujimoto, Takeshi Nakatani, Daiki Nagira, Yutaka Muto, Takehiro Hirayama, Hachiro Konaka, Masato Okada, Hyota Takamatsu, and Atsushi Kumanogoh

DOI: <https://doi.org/10.26508/lsa.202403015>

Corresponding author(s): Hyota Takamatsu, Osaka University and Atsushi Kumanogoh, Osaka University

Review Timeline:

Submission Date:	2024-08-23
Editorial Decision:	2024-11-01
Revision Received:	2025-03-31
Editorial Decision:	2025-05-21
Revision Received:	2025-05-26
Accepted:	2025-05-27

Scientific Editor: Tim Fessenden

Transaction Report:

November 1, 2024

Re: Life Science Alliance manuscript #LSA-2024-03015-T

Dr. Hyota Takamatsu
WPI Immunology Frontier Research Center
Department of Immunopathology
2-2 Yamadaoka,
Suita, Osaka 5650871
Japan

Dear Dr. Takamatsu,

Thank you for submitting your manuscript entitled "Regulator complex and lysosomal calcium release are crucial for cell migration." to Life Science Alliance. The manuscript was assessed by expert reviewers, whose comments are appended to this letter. We invite you to submit a revised manuscript addressing the Reviewer comments.

Thank you for this interesting contribution to Life Science Alliance. We are looking forward to receiving your revised manuscript.

Sincerely,

B. MANUSCRIPT ORGANIZATION AND FORMATTING:

Reviewer #1 (Comments to the Authors (Required)):

The manuscript by Jo, Kumanogoh, and colleagues reports data indicating that lysosomal calcium regulates immune cell migration by modulating the interaction between the Ragulator complex and the myosin regulator MPRIP. This follows from their prior work showing that this interaction regulates leukocyte trafficking. Notably, they completed a chemical screen and identified cardiac glycosides including ouabain as compounds that inhibit the Ragulator-MPRIP interaction. They go on to show that ouabain can suppress inflammation in mouse models of inflammatory diseases.

The central claims of the work are that 1) lysosomal calcium acts through the Ragulator-MPRIP interaction to regulate migration, 2) Ragulator localization to the lysosome is critical for its role in migration, and 3) Ouabain diminishes interaction between Ragulator and MPRIP. 4) Ouabain alleviates inflammation by suppressing immune cell recruitment.

This is interesting work with potential therapeutic implications. It provides new insights into the regulation of myosin in cell migration. However, I do have some concerns about mechanistic interpretation and technical aspects of the work.

Major points related to support for the central claims:

For the first central claim (lysosomal calcium acts through the Ragulator-MPRIP interaction to regulate migration), more data should be provided to support the conclusion that that migration is directly affected:

1.) The only assay used to monitor cell migration in this study appears to be the transwell assay. The transwell assay is heavily dependent on cell adhesion (see Belliveau et al, Nature Communications 2023). It is unclear whether the major effects of the Ragulator-MPRIP interaction are on adhesion (which is sensitive to calcium signaling) or cell movement. Either role could be relevant, as adhesion is also critical for recruitment of leukocytes from blood vessels. However, the mechanistic interpretation would be different. A direct cellular imaging assay should clarify this.

2.) Another mechanistic question that seems unresolved is whether the role of lysosomal calcium is to directly activate myosin to provide force for cell migration (which seems to be the preferred model of the authors) or to position lysosomes properly so that lysosome-associated myosin-based contraction acts in the right place to promote migration. TRPML1 is known to regulate lysosome fusion and positioning. The effects on binding between Ragulator and MPRIP could be a secondary consequence of endo-lysosomal fusion and lysosome positioning. The data in Figure 1A suggests direct regulation of the binding by calcium, but the calcium concentrations used are very high (calcium microdomains in the cytoplasm are thought to be in the 100 uM ballpark rather than millimolar) and the effects on binding are modest. The authors should discuss this question.

The second claim is adequately supported.

3.) The third claim is supported. However, Ouabain seems to have large-scale effects on the endo-lysosomal organelle system. It is not clear that all of the effects on migration and inflammation are through Lamtor1 and MPRIP. The authors may comment on this.

The fourth claim is supported.

Minor points:

1.) There are technical issues with fluorescence images throughout the figures. I assume these have to do with how the images were processed and analyzed. There are visible rectangles in most of the images that appear to be masking something out. What is being masked out? Can high resolution images without these rectangles be provided?

2.) A second technical issue with the images is that the confocal images in different channels do not seem to match perfectly. Looking closely at the images in Fig 3D, it is not clear if these images are from the same focal plane. There is a nuclear shadow in each, but the shape looks slightly different, as if the z-plane is not matched. A potential consequence of this is that it is difficult to tell whether the two proteins are on the same organelles. Both are expected to be lysosomal, and there is some overlap, but the patterns look very different. Are they really on different organelles, or is this technical imaging issue? Higher resolution

images with carefully matched focal planes should clarify this.

- 3) Multiple western blot bands including the Lamtor1 WCL bands in Fig 1A look like they have an imaging artifact in the center, perhaps saturation?
- 4.) Throughout the manuscript the approaches used are often insufficiently described. Examples include the BRET assay in Figure 1, the chemotaxis assays used, and the (presumably) cell fractionation experiment in Figure 3E. Extensive details are not needed in the text, but there should be sufficient information for the reader to understand the basis for the assay. For example, for the BRET experiment in Figure 1, what fusion proteins are used? For the migration experiments, what assay was used? Transwell?
- 5.) Fig EV1, Lamtor1 G2A localization looks largely different from LAMP1, but there is still a region with highly enriched Lamtor1 that colocalizes with Lamp1. Is Lamtor1 G2A still partly localized to endo-lysosomal organelles?
- 6.) The migration/chemotaxis phenotype for the Lamtor1 mutant could be due to secondary effects from gene expression or differentiation. Belliveau et al (2023) identified Lamtor1 (and other Ragulator components) as key regulators of HL-60 cell differentiation.
- 7.) Do the Lamtor1-KO-G2A-THP1 cells have the endogenous Lamtor1 knocked out and replaced with a G2A mutant transgene? Or was the gene edited at the endogenous loci? The authors should clarify this in the text.
- 8.) If ouabain affects Ragulator localization to lysosomes, why doesn't it affect mTORC1?
- 9.) At the bottom of page 7 "injection of alum". Is alum an abbreviation for aluminum sulfate? The compound used should be spelled out.
- 10.) For the in vivo conclusion, it would be more accurate to say leukocyte recruitment or infiltration rather than leukocyte migration, since migration was not measured directly. For example, the primary effect could be on adhesion and/or extravasation.
- 11.) A data table with the results of the chemical screens should be provided as supplementary material.

Reviewer #2 (Comments to the Authors (Required)):

In this article, Jo et al investigate TRPML1 activity on MyoII activation. In particular, the authors propose that TRPML1 activity enhances the interaction between Ragulator and MPRIP, which leads to MLC phosphatase inhibition, MyoII activation and cell migration. The article is interesting, providing evidence on the mechanism by which TRPML1 might regulate cell migration. Nonetheless, the article lacks some quantifications, and some conclusions go beyond what the figures show and must be readdressed.

Main points

- In Figure 1, the authors study the interaction between Lamtor1 and MPRIP in HEK293T cells. This is done by immunoprecipitation (IP) in buffers containing different calcium levels, finding that high calcium strengthens this interaction. While the data on IP is convincing, the image showing TRPML1 and Lamtor1 is incomplete. The picture shows a single cell with no quantification, and the impact of the treatment with the TRPML1 agonist MLSA1 or antagonist MLSI3 was not assessed. The description is also vague, "Lamtor1 and TRPML1 were located close to each other". The colocalization must be quantified as in Figures 3D and 3G. The IF is complementary to IP, because it shows that in intact cells the interaction is indeed occurring. A better IF quantification would support the NanoBRET results, used as main argument, where the differences observed are very low.
- In the migration part of figure 1, the differences observed upon MLSA1 and MLSI3 are mainly the presence of CCL2 using a transwell assay. Yet, all the experiments studying the interaction between Lamtor1 and MPRIP were done in the absence of CCL2, raising the question of whether this chemokine modulates TRPML1 activity. This aspect is not discussed in the article and needs to be clarified.
- In Figure 2, the authors use a mutant version of Lamtor1 that does not colocalize with lysosomes. In IP experiments, the interaction between Lamtor1 and MPRIP is reduced, even if this is done in extracts in which the sub-compartmentalization of the proteins is lost. This indicates that the mutant Lamtor1 itself fails in interacting with MPRIP, independently of its localization. This could indicate a dominant negative effect of the mutated protein in the pathway, explaining also the lack of interaction upon MLSA1 treatment. This is not discussed in the article and needs to be clarified.

- The identification of modulators of Lamtor1-MPRIP is interesting, but the rationale for the selection of the compounds used is absent. It would be informative for the reader to know what chemical and natural libraries were used. The description is vague and needs to be strengthened by a more precise rationale.

- In Figure 3G, the images appear to have "black squares" that delete areas of the images. Also, the cells do not have the same size. Other images present the same problem. The article's images can not be published in the current state and need to be redone.

- In general, IF images need quantifications. Fig EV1, as figure 1B, are incomplete. Overall, the article requires a set of substantial revisions before being published.

Reviewer #3 (Comments to the Authors (Required)):

This is an interesting study that addresses the role of the Ragulator complex and MPRIP. The authors show that Ragulator complex and MPRIP interact and that agonists of Ca²⁺ TRPML1 channel increase the interaction and cell migration, whereas an antagonist decreases both. The authors then explore natural compounds that may modulate the interaction, setting on ouabain as a potential regulator. After showing that ouabain actually impairs the interaction between Lamtor and MPRIP, the authors demonstrated that ouabain impairs leukocyte migration in vitro and in vivo in models of joint and lung inflammation and peritonitis.

Overall, this is an interesting and potentially important study that is very strong in some respects, and somewhat disjointed in others. The authors need to address the following:

Figure 1, the authors show that calcium increases Lamtor interaction with MPRIP. The authors should add data using A23187 (calcium ionophore) and EGTA. Does EGTA disperse the channel, Lamtor1, or both? (Fig. 1B).

Also Figure 1E, it is clear that MLSI3 impairs motility (they do not see significance in the absence of CCL2 because of showing this side-by-side with chemotaxing cells, but the trend in chemokinetic cells is as clear, just one order of magnitude smaller. Conversely, MLSA1 positive effect is only seen on chemotaxis. This needs to be discussed.

Fig EV1, Lamtor1-KO-G2A-THP1 cells were used, in which the N-terminal G2 region, a membrane tether of Lamtor1, was replaced with alanine. Although there is a reference to a previous study, this is unclear.

Figure 2, can additional calcium compensate for the lost interaction between Lamtor-KO-G2A and MPRIP? And RLC phosphorylation?

Figure 3, what is the effect of the interaction between MYPT1 and MPRIP? Does this mean that the interaction between MPRIP and LAMTOR1 or MYPT1 are mutually exclusive? Do they share a binding site?

Figure 4C, the authors address the effect of ouabain on Rac activity, but what is relevant towards myosin II activation is RhoA. What is the effect of this drug on RhoA activation?

Figure 4E/F, although joint swelling (correct the Y axis on 4E, it is JOINT) and lung injury are usually consistent with increased leukocyte migration, these parameters do not measure leukocyte migration per se. In this regard, does ouabain prevent DC activation? The authors could measure the levels of CD80/CD86 in LPS-treated DC +/- ouabain.

Figure 5, the authors show MPRIP associated to actin, and the recruitment of MYPT1 to fibers. How good is that evidence? The authors would need to include stainings of their cells in which they look at actin with MPRIP and MYPT1 to underscore these points.

Reviewer #1 (Comments to the Authors (Required)):

The manuscript by Jo, Kumanogoh, and colleagues reports data indicating that lysosomal calcium regulates immune cell migration by modulating the interaction between the Ragulator complex and the myosin regulator MPRIP. This follows from their prior work showing that this interaction regulates leukocyte trafficking. Notably, they completed a chemical screen and identified cardiac glycosides including ouabain as compounds that inhibit the Ragulator-MPRIP interaction. They go on to show that ouabain can suppress inflammation in mouse models of inflammatory diseases.

The central claims of the work are that 1) lysosomal calcium acts through the Ragulator-MPRIP interaction to regulate migration, 2) Ragulator localization to the lysosome is critical for its role in migration, and 3) Ouabain diminishes interaction between Ragulator and MPRIP. 4) Ouabain alleviates inflammation by suppressing immune cell recruitment.

This is interesting work with potential therapeutic implications. It provides new insights into the regulation of myosin in cell migration. However, I do have some concerns about mechanistic interpretation and technical aspects of the work.

Major points related to support for the central claims:

For the first central claim (lysosomal calcium acts through the Ragulator-MPRIP interaction to regulate migration), more data should be provided to support the conclusion that that migration is directly affected:

1-1.) The only assay used to monitor cell migration in this study appears to be the transwell assay. The transwell assay is heavily dependent on cell adhesion (see Belliveau et al, Nature Communications 2023). It is unclear whether the major effects of the Ragulator-MPRIP interaction are on adhesion (which is sensitive to calcium signaling) or cell movement. Either role could be relevant, as adhesion is also critical for recruitment of leukocytes from blood vessels. However, the mechanistic interpretation would be different. A direct cellular imaging assay should clarify this.

We appreciate the valuable comments that helped improve the manuscript. First, we have compared the adhesion activity of WT and Lamtor1-KO BMDCs to fibronectin-coated plates, as described in the supplementary information of our previous article (PMID: 34099704). The results showed no difference in adhesion despite the lack of Lamtor1.

Additionally, we performed adhesion assays wherein THP1 cells were seeded on VCAM-1-coated dishes for 2 h in the presence of MLSA-1 or MLSI-3. No significant difference of adhesion was observed with either treatment (Figure S1B).

Figure S1B

Regarding the effects on actomyosin-mediated migration, we have conducted additional 3D collagen environment migration assays using DMSO (negative control), MLSA-1 (TRPML1 agonist), MLSI-3 (TRPML1 antagonist), and Blebbistatin (Myosin IIA inhibitor). MLSA-1 treated cells exhibited the highest motility with quick tail contraction, whereas MLSI-3 treated cells showed decreased motility and elongated cellular morphology due to impaired rear retraction, as myosin IIA activity was inhibited by Blebbistatin (Figure 1H, Movie 1–4). These findings suggest that TRPML1 regulates cell movement via actomyosin contraction.

Figure 1H

Collectively, these findings indicate that lysosomal calcium modulates cell movement via actomyosin contraction rather than adhesion.

1-2.) Another mechanistic question that seems unresolved is whether the role of lysosomal calcium is to directly activate myosin to provide force for cell migration (which seems to be the preferred model of the authors) or to position lysosomes properly so that lysosome-associated myosin-based contraction acts in the right place to promote migration. TRPML1 is known to regulate lysosome fusion and positioning. The effects on binding between Regulator and MPRIP could be a secondary consequence of endo-lysosomal fusion and lysosome positioning. The data in Figure 1A suggests direct regulation of the binding by calcium, but the calcium concentrations used are very high (calcium microdomains in the cytoplasm are thought to be in the 100 μ M ballpark rather than millimolar) and the effects on binding are modest. The authors should discuss this question.

First, we examined lysosomes distribution during migration in 3D collagen matrices using LysoTracker-labelled BMDC with or without MLSI-3 (Figure S1C, Movie 5-6). Consistent with our previous article, a subset of lysosomes localized to the uropod of cells during migration. Notably, this uropod localization persisted even when cell retraction was impaired by MLSI-3.

Figure S1C

These results suggest that, under our experimental conditions, lysosomal positioning was not impaired by the inhibition of TRPML1-mediated Ca²⁺ efflux. Regarding lysosomal size, a previous study by Bretou et al. (2017) on TRPML1-knockout BMDCs reported no significant differences in lysosomal size with normal lysosomal trafficking. Similarly, we observed no clear differences in lysosomal size in Lamtor1-deficient cells and TRPML1 modulation. Therefore, we believed the enhanced interaction between Lamtor1 and MPRIP via lysosomal calcium is not a secondary effect of the abnormal lysosomal trafficking.

To address the concern regarding the high calcium concentration in Figure 1A, we have addressed this issue by repeating the assay with a modified protocol: the cell lysate was pretreated with EDTA for 6 h to chelate pre-existing Ca²⁺, followed by its addition to the cell lysate. Using this approach, we found that 100 μ M Ca²⁺ was sufficient to promote the binding between the Ragulator complex and MPRIP (Figure 1A). We appreciate the reviewer's comment, which helped us confirm that Lamtor1 interacted with MPRIP under physiological conditions, which strengthens our argument for a direct regulatory role of calcium in this interaction.

Figure 1A

The second claim is adequately supported.

1-3.)

The third claim is supported. However, Ouabain seems to have large-scale effects on the endo-lysosomal organelle system. It is not clear that all of the effects on migration and inflammation are through Lamtor1 and MPRIP. The authors may comment on this.

We have addressed the concern regarding ouabain's effects on the endo-lysosomal system by evaluating LC3 flux as a marker of autophagy (Figure S3E). Ouabain did not affect LC3-II protein levels. Additionally, we evaluated the effect of ouabain on LPS-induced TNF- α production by THP1 cells (Figure S4D) and found no differences in TNF- α secretion. Furthermore, ouabain treatment did not affect cell viability (Figure S3F).

Figure S3E

Figure S4D

Figure S3F

We evaluated whether ouabain suppresses cell migration in Lamtor1-KO or MPRIP-KO THP1 cells (Figure S4B), finding that ouabain did not suppress the chemotaxis upon CCL2 exposure in Lamtor1-KO and MPRIP-KO THP1 cells. This suggests that ouabain regulates cell migration in a Lamtor1 and MPRIP dependent manner.

Figure S4B

The fourth claim is supported.

Minor points:

1-4.) There are technical issues with fluorescence images throughout the figures. I assume these have to do with how the images were processed and analyzed. There are visible rectangles in most of the images that appear to be masking something out. What is being masked out? Can high resolution images without these rectangles be provided?

We apologize for any confusion caused by the insertion of rectangles and the low resolution of the fluorescence images. To address this, we have repeated the imaging and quantitative analysis with minor modifications to the staining protocol. The new figures clearly show the relevant cellular structures and protein localizations. Specifically, we have revised the figures of colocalization of Lamtor1 and TRPML1 in Fig 1B; colocalization of Lamtor1 and Lamp1 with or without ouabain in Figure 3D; localization of Lamtor1 and Lamp1 in ATP1A1-KD THP1 cells with or without ouabain in Figure 3G; and colocalization of Lamtor1 and Lamp1 in Lamtor1-KO-G2A-THP1 cells in Figure S2A.

Fig 1B

Fig 3D

Fig 3G

Fig S2A

2.) A second technical issue with the images is that the confocal images in different channels do not seem to match perfectly. Looking closely at the images in Fig 3D, it is not clear if these images are from the same focal plane. There is a nuclear shadow in each, but the shape looks slightly different, as if the z-plane is not matched. A potential consequence of this is that it is difficult to tell whether the two proteins are on the same organelles. Both are expected to be lysosomal, and there is some overlap, but the patterns look very different. Are they really on different organelles, or is this technical imaging issue? Higher resolution images with carefully matched focal planes should clarify this.

We apologize for any confusion caused by the previous images and thank the reviewer for bringing this to our attention. As mentioned in the previous response, the revised figure provides a more accurate representation of the protein localization (Figure 3D), confirming that Lamtor1 is present on lysosomal organelles.

Fig 3D

3) Multiple western blot bands including the Lamtor1 WCL bands in Fig 1A look like they have an imaging artifact in the center, perhaps saturation?

We acknowledge the potential imaging artifact of the Lamtor1-Flag WCL bands in Figure 1A and MPRIP-V5 bands in the IP of Figure 3B. To address this concern, we have conducted new western blot experiments. We have presented more accurate and quantifiable results in the revised manuscript, allowing a better interpretation of protein expression levels.

Figure 1A

Figure 3B

4.) Throughout the manuscript the approaches used are often insufficiently described. Examples include the BRET assay in Figure 1, the chemotaxis assays used, and the (presumably) cell fractionation experiment in Figure 3E. Extensive details are not needed in the text, but there should be sufficient information for the reader to understand the basis for the assay. For example, for the BRET experiment in Figure 1, what fusion proteins are used? For the migration experiments, what assay was used? Transwell?

We appreciate your feedback regarding the insufficient description of our experimental methods. We have revised the manuscript to include more detailed methodological information, enabling readers to better understand the assay procedures.

5.) Fig S1, Lamtor1 G2A localization looks largely different from LAMP1, but there is still a region with highly enriched Lamtor1 that colocalizes with Lamp1. Is Lamtor1 G2A still partly localized to endo-lysosomal organelles?

We appreciate the reviewer's comment regarding the localization of Lamtor1 G2A. In Lamtor1-KO THP1 cells with Lamtor1-G2A restored, the mutant was diffusely distributed throughout the cytosol, confirming that the N-terminal lipid modification is necessary for lysosomal membrane anchoring. We have revised Figure S2A as shown in the image below.

Figure S2A

6.) The migration/chemotaxis phenotype for the Lamtor1 mutant could be due to secondary effects from gene expression or differentiation. Belliveau et al (2023) identified Lamtor1 (and other Ragulator components) as key regulators of HL-60 cell differentiation.

Thank you for raising this important point regarding potential secondary effects of Lamtor1 mutation on gene expression or differentiation influencing the migration/chemotaxis phenotype. To address this concern, we have conducted RNA sequencing (RNA-seq) analysis to compare Lamtor1-KO-THP1 cells with Lamtor1 or Lamtor1-G2A mutant restored. The expression of gene sets related to migration, components of the myosin light chain phosphatase (MLCP) complex, and chemokine receptors did not differ between the two cell lines. These findings strengthen our conclusion that Lamtor1 directly regulates cell migration and chemotaxis, independent of major changes in the expression of migration-related genes.

7.) Do the Lamtor1-KO-G2A-THP1 cells have the endogenous Lamtor1 knocked out and replaced with a G2A mutant transgene? Or was the gene edited at the endogenous loci? The authors should clarify this in the text.

We apologize for any lack of clarity regarding the Lamtor1-KO-G2A-THP1 cells. These cells were generated by expressing the Lamtor1-G2A mutant through the transient administration of gRNA in stably Cas9-expressing THP1 cells, as described in our previous manuscript (Nakatani et al. PMID: 34099704). As a control, the expression of the full-length Lamtor1 rescued the Lamtor1-KO THP1 cells. Therefore, there is no concern regarding the dominant-negative effects of endogenous Lamtor1. We have added this information to the relevant sections of the revised manuscript to ensure readers clearly understand how these cells were generated.

8.) If ouabain affects Regulator localization to lysosomes, why doesn't it affect mTORC1?

We acknowledge the valid concern that mTORC1 activity was affected by the dislocation of Lamtor1 from lysosomes. We assessed the phosphorylation of ppS6K, a downstream target of mTORC1, in the presence of ouabain. The phosphorylation of ppS6K was not suppressed when cells were treated within 12 h of treatment. However, after 24 to 48 h, the phosphorylation of ppS6K was gradually reduced. These findings suggest that while prolonged ouabain exposure may affect multiple cellular targets or signaling pathways, mTORC1 signaling remains largely unaffected. This information has been added to the revised manuscript (Figure S3D).

Figure S3D

9.) At the bottom of page 7 "injection of alum". Is alum an abbreviation for aluminum sulfate? The compound used should be spelled out.

We have added “aluminum hydroxide,” the full name of the compound, in the revised manuscript where it first appears.

10.) For the in vivo conclusion, it would be more accurate to say leukocyte recruitment or infiltration rather than leukocyte migration, since migration was not measured directly. For example, the primary effect could be on adhesion and/or extravasation.

We have amended “leukocyte recruitment” to “leukocyte migration” where appropriate.

11.) A data table with the results of the chemical screens should be provided as supplementary material.

Per your recommendation, we have included a comprehensive table with the results of the chemical screens as supplementary material.

Reviewer #2 (Comments to the Authors (Required)):

In this article, Jo et al investigate TRPML1 activity on MyoII activation. In particular, the authors propose that TRPML1 activity enhances the interaction between Ragulator and MPRIP, which leads to MLC phosphatase inhibition, MyoII activation and cell migration. The article is interesting, providing evidence on the mechanism by which TRPML1 might regulate cell migration. Nonetheless, the article lacks some quantifications, and some conclusions go beyond what the figures show and must be readdressed.

Main points

- In Figure 1, the authors study the interaction between Lamtor1 and MPRIP in HEK293T cells. This is done by immunoprecipitation (IP) in buffers containing different calcium levels, finding that high calcium strengthens this interaction. While the data on IP is convincing, the image showing TRPML1 and Lamtor1 is incomplete. The picture shows a single cell with no quantification, and the impact of the treatment with the TRPML1 agonist MLSA1 or antagonist MLSI3 was not assessed. The description is also vague, "Lamtor1 and TRPML1 were located close to each other". The colocalization must be quantified as in Figures 3D and 3G. The IF is complementary to IP, because it shows that in intact cells the interaction is indeed occurring. A better IF quantification would support the NanoBRET results, used as main argument, where the differences observed are very low.

We appreciate your feedback and have addressed your concerns by conducting additional experiments and analyses. We have constructed a TRPML1-V5 plasmid and performed co-immunoprecipitations with Lamtor1-Flag, which confirmed the interaction of Lamtor1 and TRPML1 (Figure 1C). Confocal microscopic analysis further demonstrated that TRPML1 co-localizes with Lamtor1 in the lysosomes. Quantitative analysis of their colocalization is shown in Figure 1B. We believe these results strengthen our study and address the reviewer's concern.

Figure 1C

Figure 1B

- In the migration part of figure 1, the differences observed upon MLSA1 and MLSI3 are mainly the presence of CCL2 using a transwell assay. Yet, all the experiments studying the interaction between Lamtor1 and MPRIP were done in the absence of CCL2, raising the question of whether this chemokine modulates TRPML1 activity. This aspect is not discussed in the article and needs to be clarified.

In response to your concern, we evaluated TRPML1 expression levels following CCL2 treatment, finding that it did not significantly alter TRPML1 protein expression.

Although TRPML1 induced the binding of Lamtor1 to MPRIP and the phosphorylation of MLC even without CCL2, cell migration was not affected by TRPML1 modulators in the absence of chemokines. Cell migration requires the coordinated balance between chemokine-mediated protrusion of the leading edge and Myosin IIA-mediated contraction of the trailing edge. Additionally, at the rest, lysosomes preferentially localize at the perinuclear region, whereas upon chemokine stimulation lysosomes redistribute to the uropod where TRPML1-mediated Lamtor1 and MPRIP interaction occur. Therefore, it is probable that cells do not move forward in the absence of the chemokine signaling, despite myosin IIA activation, and that chemokines are responsible for the spatiotemporal localization of lysosomes wherein Lamtor1 and MPRIP induce cell movement.

- In Figure 2, the authors use a mutant version of Lamtor1 that does not colocalize with lysosomes. In IP experiments, the interaction between Lamtor1 and MPRIP is reduced, even if this is done in extracts in which the sub-compartmentalization of the proteins is lost. This indicates that the mutant Lamtor1 itself fails in interacting with MPRIP, independently of its localization. This could indicate a dominant negative effect of the mutated protein in the pathway, explaining also the lack of interaction upon MLSA1 treatment. This is not discussed in the article and needs to be clarified.

Thank you for your insightful comment. We have conducted a co-immunoprecipitation assay between Lamtor1-G2A-Flag and MPRIP-V5. The interaction between these proteins was less than that of Lamtor1-Full-Flag and MPRIP-V5. Additionally, we evaluated whether the Lamtor1-G2A-Flag and MPRIP-V5 interaction increased by the presence of Ca²⁺ in the cell lysate. Unexpectedly, Ca²⁺ administration did not enhance the interaction of these proteins. Based on these results, we propose two explanations for the reduced interaction between Lamtor1-G2A and MPRIP. First, as you suggested, the G2A mutation may alter the protein's structure or its binding properties, even though the components of the Ragulator complex, including Lamtor2-5, were pulled down with Lamtor1-G2A as shown in Figure S3G. Indeed, we did observe a slight shift of Lamtor1-G2A band size in western blotting data (Figure 2A, 2D), suggesting the potential aggregation or structural alterations that could affect Lamtor1 and MPRIP binding.

Figure S3G

Figure 2A

Figure 2D

Another possible explanation is that lysosomal factors other than TRPML1, including lipids of the lysosomal membrane and scaffold proteins, increase Lamtor1-MPRIP interaction. As these points were not addressed in this manuscript, future work will explore these possibilities.

Regarding the potential dominant negative effect, we generated Lamtor1-KO-G2A-THP1 cells by expressing the Lamtor1-G2A mutant through the transient administration of gRNA in stably Cas9-expressing THP1 cells, as described in our previous manuscript (Nakatani et al. PMID: 34099704). As a control, the expression of the full-length Lamtor1 rescued the Lamtor1-KO THP1 cells. Therefore, there is no concern regarding the dominant-negative effect of endogenous Lamtor1.

- The identification of modulators of Lamtor1-MPRIP is interesting, but the rationale for the selection of the compounds used is absent. It would be informative for the reader to know what chemical and natural libraries were used. The description is vague and needs to be strengthened by a more precise rationale.

Regarding the selection of modulators for the Lamtor1-MPRIP interaction, we acknowledge that our initial description was insufficient. We have used the Prestwick chemical and phytochemical library of substances clinically approved by the FDA. This information has been added to the Results and Material and Method sections in the revised manuscript.

- In Figure 3G, the images appear to have "black squares" that delete areas of the images. Also, the cells do not have the same size. Other images present the same problem. The article's images can not be published in the current state and need to be redone.

We sincerely apologize for the image quality issues in the Figures, including Figure 3G. We have repeated the imaging of high-resolution images and carefully reviewed all images in the revised manuscript.

Fig 1B

Fig 3D

Fig S2A

Fig 3G

- In general, IF images need quantifications. Fig S1, as figure 1B, are incomplete. Overall, the article requires a set substantial revisions before being published.

We appreciate your comments regarding the need for quantification of immunofluorescence (IF) images. In response, we have conducted comprehensive quantitative analyses for all IF images, including those in Figure S1 (Figure S2A in the revised manuscript) and Figure 1B. These quantifications provide a more robust and objective representation of our results.

Fig 1B

Fig S2A

Reviewer #3 (Comments to the Authors (Required)):

This is an interesting study that addresses the role of the Ragulator complex and MPRIP. The authors show that Ragulator complex and MPRIP interact and that agonists of Ca²⁺ TRPML1 channel increases the interaction and cell migration, whereas an antagonist decreases both. The authors then explore natural compounds that may modulate the interaction, setting on ouabain as a potential regulator. After showing that ouabain actually impairs the interaction between Lamtor and MPRIP, the authors demonstrated that ouabain impairs leukocyte migration in vitro and in vivo in models of joint and lung inflammation and peritonitis.

Overall, this is an interesting and potentially important study that is very strong in some respects, and somewhat disjointed in others. The authors need to address the following:

Figure 1, the authors show that calcium increases Lamtor interaction with MPRIP. The authors should add data using A23187 (calcium ionophore) and EGTA. Does EGTA disperse the channel, Lamtor1, or both? (Figure 1B).

We evaluated the interaction between Lamtor1 and MPRIP in HEK293T cells expressing Lamtor1-Flag and MPRIP-V5 treated with A23187, wherein Lamtor1-MPRIP binding levels were not statistically higher than those in the cells treated with TRPML1 agonists (Figure S1A). This suggests that the lysosome-derived calcium, rather than a diffuse intracellular calcium increase, may promote Lamtor1-MPRIP binding.

Figure S1A

Furthermore, we have added EGTA to the cell lysate of Lamtor1-Flag- and MPRIP-V5-expressing cells, which inhibited the calcium-induced enhancement of Lamtor1-MPRIP interaction (Figure 1A).

Figure 1A

Also Figure 1E, it is clear that MLSI3 impairs motility (they do not see significance in the absence of CCL2 because of showing this side-by-side with chemotaxing cells, but the trend in chemokinetic cells is as clear, just one order of magnitude smaller. Conversely, MLSA1 positive effect is only seen on chemotaxis. This needs to be discussed.

Thank you for your insightful comment. In our previous article, we found differences in the role of MLSA-1 and MLSI-3 in chemotaxis in the absence of CCL2. However, after repeated experiments, we concluded that neither MLSA-1 nor MLSI-3 has a statistically significant effect on cell motility in the absence of CCL2 (Figure 1F).

Figure 1F

Cell migration requires a coordinated balance between chemokine-mediated protrusion at the leading edge and Myosin IIA-mediated contraction at the trailing edge. At rest, lysosomes preferentially localize at the perinuclear region. However, upon chemokine stimulation, lysosomes redistribute to the uropod where TRPML1-mediated Lamtor1 and MPRIP interaction occur. Therefore, Ca²⁺ modulation via TRPML1 may more effectively regulate cell motility in the presence of CCL2 than under resting conditions.

Fig S1, Lamtor1-KO-G2A-THP1 cells were used, in which the N-terminal G2 region, a membrane tether of Lamtor1, was replaced with alanine. Although there is a reference to a previous study, this is unclear.

We apologize for the poor explanation of Lamtor1-KO-G2A-THP1 cells. We generated Lamtor1-KO-G2A-THP1 cells by expressing the Lamtor1-G2A mutant through the transient administration of gRNA in stably Cas9-expressing THP1 cells. As a control, the expression of full-length Lamtor1 rescued the Lamtor1-KO THP1 cells. Therefore, there is no concern regarding the dominant-negative effect of endogenous Lamtor1. We have added the corresponding references of these details in the revised manuscript (Nakatani et al, PMID: 34099704; Tsujimoto et al, PMID: 36444797).

Figure 2, can additional calcium compensate for the lost interaction between Lamtor-KO-G2A and MRIP? And RLC phosphorylation?

Thank you for your insightful comment. We have conducted a co-immunoprecipitation assay between Lamtor1-G2A-Flag and MPRIP-V5. The interaction between two proteins was less than that of Lamtor1-Full-Flag and MPRIP-V5. Additionally, we evaluated whether the Lamtor1-G2A-Flag and MPRIP-V5 interaction increased by the presence of Ca²⁺ in the cell lysate. Unexpectedly, Ca²⁺ administration did not enhance the interaction of these proteins.

Based on these results, we propose two explanations for the reduced interaction between Lamtor1-G2A and MPRIP. First, as you suggested, the G2A mutation may have alter the protein's structure or its binding properties, even though the components of the Ragulator complex, including Lamtor2-5, were pulled down with Lamtor1-G2A as shown in Figure S3G. Indeed, we did observe a slight shift of Lamtor1-G2A band size in western blotting data (Figure 2A, 2D), suggesting the potential aggregation or structural alterations that could affect Lamtor1 and MPRIP binding. Another possible explanation is that lysosomal factors other than TRPML1, including lipids of the lysosomal membrane and scaffold proteins, increase Lamtor1-MPRIP interaction. As these points were not addressed in this manuscript, future work will explore these possibilities.

Figure S3G

Figure 2A

Figure 2D

Regarding MLC phosphorylation levels in the context of defective localization of Lamtor1 to lysosomes, their levels in Lamtor1-KO-G2A THP1 cells were less than those of Lamtor1-KO-Full THP1 cells (Figure 2C). Additionally, MLSA-1 treatment did not increase MLC phosphorylation in Lamtor1-KO-G2A-THP1 cells (Figure 2F). Therefore, we believe that the place of Lamtor1 in lysosomes is important for TRPML1-mediated actomyosin contraction.

Figure 2C

Figure 2F

Figure 3, what is the effect of the interaction between MYPT1 and MPRIP? Does this mean that the interaction between MPRIP and LAMTOR1 or MYPT1 are mutually exclusive? Do they share a binding site?

The binding regions of MPRIP for MYPT1 or Lamtor1 differ. MYPT1 binds to the C-terminal region of MPRIP (amino acids 824-879), whereas Lamtor1 binds to the N-terminal region (amino acids 1-150). We previously showed that Lamtor1 interaction with MPRIP dissociates MYPT1 from MPRIP, which is particular since MYPT1 and Lamtor1 do not share the same binding site on MPRIP. We thus considered that Lamtor1-MPRIP interaction may induce conformational changes of MPRIP that hinders MYPT1 binding. Although we did not address this mechanism in the manuscript, future studies to characterize this three-protein-interaction are warranted.

Figure 4C, the authors address the effect of ouabain on Rac activity, but what is relevant towards myosin II activation is RhoA. What is the effect of this drug on RhoA activation?

We evaluated the RhoA activity in THP1 cells with or without ouabain. GTP-bound RhoA was pulled down with GST-Rhotekin-RBD and detected by western blotting with an anti-mouse RhoA antibody. GST-RhoA levels did not differ, even at high concentrations of ouabain (Figure 4C).

Figure 4C

Figure 4E/F, although joint swelling (correct the Y axis on 4E, it is JOINT) and lung injury are usually consistent with increased leukocyte migration, these parameters do not measure leukocyte migration per se. In this regard, does ouabain prevent DC activation? The authors could measure the levels of CD80/CD86 in LPS-treated DC +/- ouabain.

We appreciated the helpful comment. We evaluated the expression levels of CD80 and CD86 in BMDCs stimulated with LPS for 6 and 24 h with or without ouabain treatment. Ouabain did not change the expression of CD80 and CD86 at any time point. Additionally, ouabain did not suppress TNF- α production in LPS-stimulated THP1 cells. These results suggest that ouabain regulates cell recruitment rather than DC activation and pro-inflammatory cytokine production.

Figure S4E

Figure 5, the authors show MPRIP associated to actin, and the recruitment of MYPT1 to fibers. How good is that evidence? The authors would need to include stainings of their cells in which they look at actin with MPRIP and MYPT1 to underscore these points.

We appreciated the valuable suggestion. To investigate this, we examined the localization of MYPT1 and phalloidin-stained F-actin in HEK293T cells with or without MPRIP-V5 overexpression. We found that MYPT1 preferentially localized to the phalloidin-positive regions when cells overexpressed MPRIP-V5. On the contrary, MYPT1 was diffusely distributed in the cytoplasm of HEK293T cells that expressed slightly less MPRIP-V5. These results confirmed that MPRIP increased the presence of MYPT1 in the F-actin bundles.

May 21, 2025

RE: Life Science Alliance Manuscript #LSA-2024-03015-TR

Dr. Hyota Takamatsu
Osaka University
Department of Immunopathology
2-2 Yamadaoka,
Suita, Osaka 5650871
Japan

Dear Dr. Takamatsu,

Thank you for submitting your revised manuscript entitled "The Ragulator complex and lysosomal calcium release are crucial for cell migration". Your manuscript was returned to original Referees 1 and 2 for evaluation. As you will see both reviewers commend the significant improvements made to this work and recommend publication. We invite you to consider the minor points raised by Reviewer 1. We would be happy to publish your paper in Life Science Alliance pending these and final revisions necessary to meet our formatting guidelines.

- Please add ORCID ID for corresponding author -- you should have received instructions on how to do so.
- Please add the X and Bluesky handles of your host institute/organization as well as your own or/and one of the authors in our system.
- Figure 5 is a graphical abstract. Please upload it with the file designation "Graphical Abstract", not as a separate figure 5, and remove its legend from the manuscript file and call-out from the manuscript text. The graphical abstract cannot be duplicated as the figure from the manuscript.
- Please be sure that all authors are mentioned in the authors' contributions section in the manuscript file.
- The contributions selected for Atsushi Kumanogoh do not qualify them for authorship. Please either update the contributions in our system and the Author Contributions section of the manuscript or let us know if the author needs to be removed (and added eventually to the acknowledgment section).
- LSA allows supplementary figures but not EV Figures; please update labels for the Supplementary Figures, Fig EV1A = Fig S1A).
- There is a call-out for Figure S4H, and this figure doesn't have this panel -- please correct.
- Please add callouts for Figures 2A; S3A-C; S4E and all movies to your main manuscript text.
- Please add molecular weight markers to all protein blots in the Supplemental Figures.
- Please provide the full details on RNA seq performed by the Genome Information Research Center.
- Please note the inclusion of RNA seq data files in the Data Availability section.

A. FINAL FILES:

B. MANUSCRIPT ORGANIZATION AND FORMATTING:

Sincerely,

Reviewer #1 (Comments to the Authors (Required)):

The revised manuscript by Jo et al addresses all of my major concerns. They have added an impressive amount of new data to address the questions raised by the reviewers. The RNA-seq data is an especially nice addition. I have a few minor points that should be addressed in the final version.

1.) In Fig 1H, the morphology of cells treated with MLSI-3 looks wildly different from control cells and also very different from blebbistatin treated cells. In the text, the authors describe the morphology of these cells as "resembling that of Blebbistatin-treated cells". This is not consistent with the images provided. The MLSI-3 treated cells have a complicated, branched morphology that suggests an inability to retract former extensions and maintain simple polarized morphology. The blebbistatin-treated cells are elongated, but they do not show the kind of branching observed for the MLSI-3 treated cells. The authors should rewrite their description of these morphologies.

2.) In Fig 4C, the y-axis label is incorrect.

3.) I have some concern about the quantification of the Western blots in Fig 4C. By eye, the band for GTP-RhoA with 100 nM Ouabain looks smaller than the bands for 0 and 50. The RhoA for normalization for this band looks to be stronger or larger than the corresponding bands for 0 and 50. However, in the quantification, all values are very close to 1. Also, the band for GDP looks dramatically weaker than the other bands. Yet, this is still quantified as ~0.5. I wonder if saturation of the bands or inaccurate background subtraction led to smaller than appropriate normalized values?

Reviewer #2 (Comments to the Authors (Required)):

Dear all,

I have reviewed the newest version of the manuscript, that includes a significant amount of new data and corrections/quantifications as compared to the initial submission. In particular, the quality of the immunofluorescence images greatly improved, which together with the quantifications make the article more convincing.

The text also evolved in agreement with the revision, making it more complete than the initial manuscript. It includes more methodological details, and a better description of the compounds used in the screening of modulators of the Regulator complex/MPRIP interaction.

The questions raised during the first revision were conveniently addressed. Notably the interaction between TRPML1 and Lamtor by co-IP and the immunofluorescence images and quantification.

I think that the actual version of this article can make a contribution to the field of cell migration by providing additional mechanistic insight on the role of lysosomes in the control of leukocyte motility.

1.) In Fig 1H, the morphology of cells treated with MLSI-3 looks wildly different from control cells and also very different from blebbistatin treated cells. In the text, the authors describe the morphology of these cells as "resembling that of Blebbistatin-treated cells". This is not consistent with the images provided. The MLSI-3 treated cells have a complicated, branched morphology that suggests an inability to retract former extensions and maintain simple polarized morphology. The blebbistatin-treated cells are elongated, but they do not show the kind of branching observed for the MLSI-3 treated cells. The authors should rewrite their description of these morphologies.

As your suggestion, we have revised the text accordingly.

2.) In Fig 4C, the y-axis label is incorrect.

We have corrected the figure axis.

2.) I have some concern about the quantification of the Western blots in Fig 4C. By eye, the band for GTP-RhoA with 100 nM Ouabain looks smaller than the bands for 0 and 50. The RhoA for normalization for this band looks to be stronger or larger than the corresponding bands for 0 and 50. However, in the quantification, all values are very close to 1. Also, the band for GDP looks dramatically weaker than the other bands. Yet, this is still quantified as ~0.5. I wonder if saturation of the bands or inaccurate background subtraction led to smaller than appropriate normalized values.

Thank you for your precise observation of our quantification in Fig 4C. We reanalyzed the signals with careful background subtraction and normalization to RhoA-Total signals (As negative controls, GDP was applied to THP1 cells, and we calculated the RhoA-GTP ratio relative to the total RhoA in THP1 cells not treated with ouabain). As a result, we found that

1) although some fluctuations were observed in the data, no decrease in RhoA-GTP was observed even when the dose of ouabain was increased; and 2) as you noted, the ratio of RhoA-GTP in GDP administration group was lower than in our previous data (0.50 ± 0.07 vs 0.29 ± 0.04 , mean \pm SD). We have revised our manuscript to include the updated quantification figure. Thank you again for your insightful feedback.

May 27, 2025

RE: Life Science Alliance Manuscript #LSA-2024-03015-TRR

Dr. Hyota Takamatsu
Osaka University
Department of Immunopathology
2-2 Yamadaoka,
Suita, Osaka 5650871
Japan

Dear Dr. Takamatsu,

Thank you for submitting your Research Article entitled "The Ragulator complex and lysosomal calcium release are crucial for cell migration.". It is a pleasure to let you know that your manuscript is now accepted for publication in Life Science Alliance. Congratulations on this interesting work.

DISTRIBUTION OF MATERIALS:

Again, congratulations on a very nice paper. I hope you found the review process to be constructive and are pleased with how the manuscript was handled editorially. We look forward to future exciting submissions from your lab.

Sincerely,
